ecology, biological applications

trophic cascade, alternative stable states, risk landscapes, management

**Author for correspondence:**
Kristy J. Kroeker
e-mail: kkroeker@ucsc.edu

†Dr Hoshijima passed away on 7 August 2019.

# Southeast Alaskan kelp forests: inferences of process from large-scale patterns of variation in space and time

Torrey R. Gorra, Sabrina C. R. Garcia, Michael R. Langhans, Umihiko Hoshijima†, James A. Estes, Pete T. Raimondi, M. Tim Tinker, Michael C. Kenner and Kristy J. Kroeker

Department of Ecology and Evolutionary Biology, University of California, Santa Cruz, CA, USA

KJK, 0000-0002-5766-1999

Humans were considered external drivers in much foundational ecological research. A recognition that humans are embedded in the complex inter-action networks we study can provide new insight into our ecological paradigms. Here, we use time-series data spanning three decades to explore the effects of human harvesting on otter–urchin–kelp trophic cascades in southeast Alaska. These effects were inferred from variation in sea urchin and kelp abundance following the post fur trade repatriation of otters and a subsequent localized reduction of otters by human harvest in one location. In an example of a classic trophic cascade, otter repatriation was followed by a 99% reduction in urchin biomass density and a greater than 99% increase in kelp density region wide. Recent spatially concentrated harvesting of otters was associated with a localized 70% decline in otter abundance in one location, with urchins increasing and kelps declining in accordance with the spatial pattern of otter occupancy within that region. While the otter–urchin–kelp trophic cascade has been associated with alternative community states at the regional scale, this research highlights how small-scale variability in otter occupancy, ostensibly due to spatial variability in harvesting or the risk landscape for otters, can result in within-region patchiness in these community states.

## 1. Background

Despite increasing attention paid to the ecological role that humans play in ecosystems, our understanding of how human behaviours influence well-known ecological paradigms is still limited [1,2]. The sea otter–sea urchin–kelp trophic cascade is one of the most well known of these ecological paradigms [3]. At the broadest level, our understanding of this trophic cascade is based on the presence/absence of sea otters (*Enhydra lutris*) in an ecosystem, linked to human exploitation patterns associated with the maritime fur trade and subsequent repatriation patterns in the North Pacific [4]. More recent work has illustrated the effect of sea otters on kelp forest community structure via space-for-time comparisons of locations differing in the duration of otter occupancy post re-introductions [5–7]. While this conceptual framework includes human impacts *on* the ecosystem via intensive harvesting or reintroduction of otters, it does not adequately capture the more nuanced role humans can play *in* the ecosystem where otters and humans co-occur and interact. For example, indigenous communities coexisted with sea otters for thousands of years prior to the maritime fur trade [8]. Food web models including human hunter–gatherers suggest humans acted as generalists and could have promoted the resilience of the ecosystem by prey-switching as resources fluctuated through time [9]. More specifically, archaeological evidence suggests

that humans had access to both otters and abundant shellfish [10,11]. This finding runs counter to our understanding of a sea otter-driven trophic cascade, whereby the presence of otters is associated with stark reductions in local shellfish populations. This evidence raises important, new questions about the role of humans in modern marine food webs regarding the conditions under which predators and prey can coexist in ecosystems typified by strong trophic cascades and alternative stable states.

Sea otters are exemplary keystone predators [12], the influence of which occurs via a trophic cascade from predatory sea otters to herbivorous sea urchins (one of the otter's prey) to kelp and other macroalgae (the urchins' prey). Kelp forests, in turn, have a broad array of knock-on effects (sensu [13]) on other species and ecological processes [14]. Sea otters are also voracious predators of other shellfish, including abalone, mussels and clams [15–17]. The negative direct effect of sea otters on their macroinvertebrate prey can manifest as human costs because the sea otters' macroinvertebrate prey base is also the foundation for several commercial, subsistence and recreational shellfisheries. In contrast, the positive indirect effects of sea otters on kelp commonly manifest as human benefits because kelp forests provide numerous ecosystem services, including habitat provisioning for other species, carbon sequestration and wave attenuation, among others [5,18–20].

Sea otters were exterminated from southeast Alaska during the maritime fur trade, then reintroduced into this area in the late 1960s [21]. With protection under the Marine Mammal Protection Act in the United States, populations have spread and grown [22], although harvest by indigenous communities is allowed and has occurred in some areas [23]. As sea otters have recovered, the resulting loss of local shellfisheries has led to resource conflicts and a call by local communities for the management of sea otter populations [23,24]. However, any plan for natural resource management through the limitation of sea otters raises several further questions, including how reductions in sea otter population densities would affect other ecosystem services provisioned by kelp forests. A better understanding of the interactions between humans, sea otters, urchins and kelp, and the spatial scale over which humans influence sea otter behaviour and abundance, may provide insight into opportunities for co-management of sea otters, kelp forests and shellfisheries.

While previous research has documented distinct, alternative ecosystem states associated with otter presence and absence across broad geographies [4,25], smaller-scale spatial variation in habitat usage by sea otters within regions associated with human activity provides an opportunity to further explore and elucidate the conditions over which kelp forests and productive shellfisheries may be able to co-occur. For example, spatial variability in predation pressure by humans or landscapes of fear for sea otters, in which spatial variation in predation risk influences otters' behaviour and distribution [26], could potentially affect otters' ecological effects at spatial scales smaller than previously recognized, even in ecosystems typically characterized by alternate stable states at larger spatial scales. Here, we use time series in two regions of southeast Alaska spanning three decades to highlight the functional relationships between humans, sea otters, urchins and kelp created by within-region spatial variability in otter populations.

## 2. Methods

### (a) System and study design

Our study was done in two areas of southeast Alaska—Torch Bay and Sitka Sound (figure 1). Subtidal reefs were initially sampled at both areas in 1988. Reintroduced sea otters had recolonized Torch Bay by about 1986 but remained rare in Sitka Sound at the time of these initial surveys [4]. Torch Bay was resurveyed in 2003 and again in 2019, at which times otters were at or near carrying capacity (fig. 3 from [22]). Sitka Sound was resurveyed in 2009, at which time otters were abundant and widespread in the area [25,27], and again in 2018, following a period of intensive sea otter harvest and population reduction (particularly near the town of Sitka) [3,23].

### (b) Sea otter surveys

Sea otter populations in southeast Alaska have been surveyed intermittently since the early 1970s. Tinker et al. [22] used these data in conjunction with a Bayesian state model to project area-specific trends in abundance relative to estimated carrying capacity. One such area was Sitka Sound (N05 in fig. 1 from [22]). We use these data (fig. 5, panel B from [22]) to characterize the trend in sea otter abundance in Sitka Sound, and the further analyses of Raymond et al. [23] to estimate the influence of Native harvest on this local sub-population (fig. 3, panel B from [23]). Although exact harvest locations were not reported, the hunters did report that they endeavoured to minimize their travel distances, resulting in an inverse relationship between harvest intensity and distance from population centres of hunters [23]. While sea otter density in Torch Bay remained relatively low throughout the study period, there has been no known harvest (Torch Bay occurs within the confines of Glacier Bay National Park), and the local population is thought to have been at or near carrying capacity since the late 1980s (fig. 3 from [22]).

In February 2018, we conducted surveys to determine the relative abundance of sea otters at each subtidal sample site in Sitka Sound (see below) to infer the spatial influence of human activity or harvesting on the distribution of sea otters. We did this prior to the habitat surveys (undertaken in August 2018, see below) out of concern that the more intensive boating that occurs during summer months in Sitka Sound would affect otter presence and detection. For each site assessment, we anchored the boat at the site, and three observers searched for otters with binoculars for 1 min, followed by a 4 min rest period. We then repeated this sampling protocol two more times with a 4 min break in between each survey. The search area (360 degrees around our boat) was divided into three exhaustive and mutually exclusive sectors, each counted by a dedicated observer. Observations occurred over a 10-day window from 8 to 18 February 2018 from 10.00 to 15.00. Weather conditions ranged from sunny to overcast. Observers recorded the number and geographic coordinates of all otters observed. We used these combined counts as a spatial index for the abundance of otters at each sample site. The spatial index was developed from a logistic regression using latitude and longitude as predictor variables and otter presence (1) or absence (0) as the response. Hence, the fitted surface represented the probability of the presence of at least one otter as a function of geographic location.

### (c) Subtidal community surveys

Habitat and sea urchin sampling methods are described in detail by Estes & Duggins [4]. Sites were initially chosen as a random sample of shoreline intersections of a grid superimposed over a navigational chart ($n = 11$ for Torch Bay; $n = 22$ for Sitka Sound). The spatial extent of both sample areas was determined by the maximum distance that could be safely travelled from the

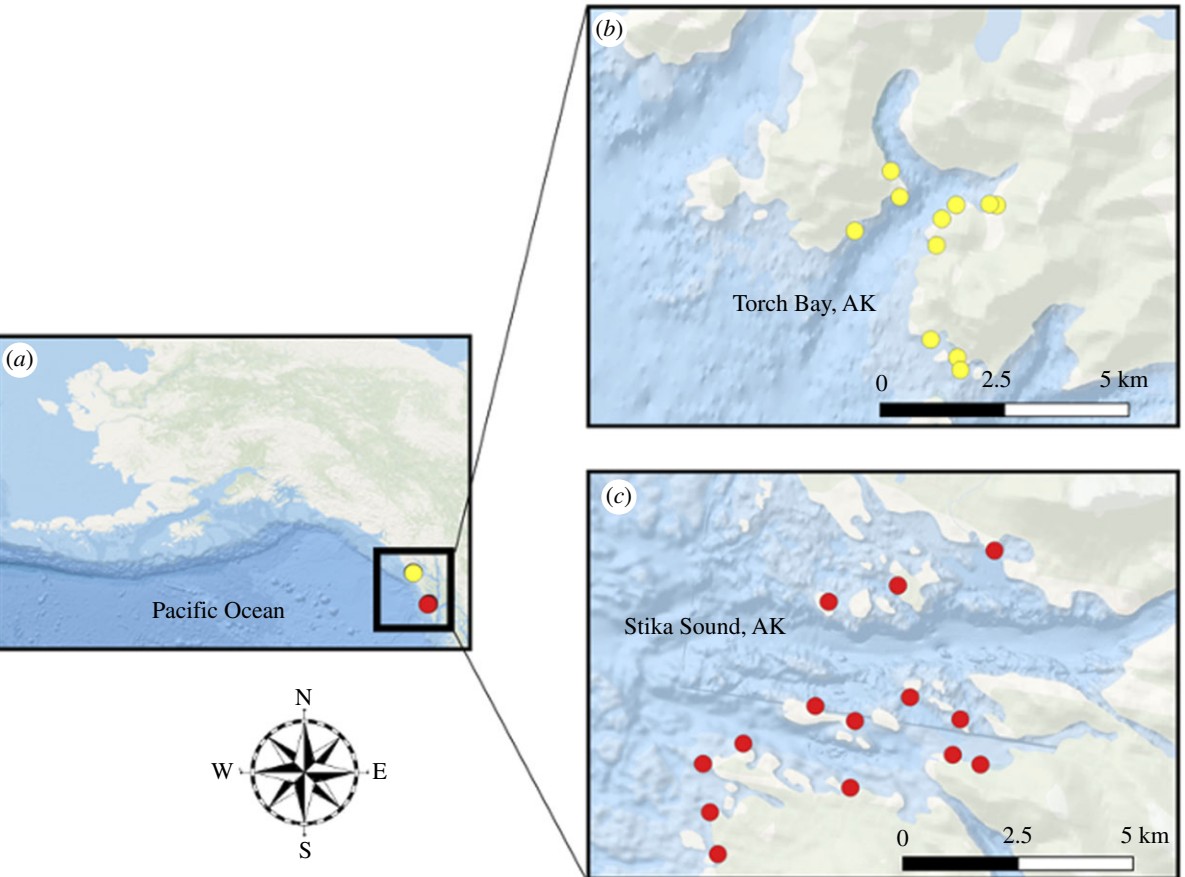

**Figure 1.** Map of study area showing location of Torch Bay (northern site, yellow) and Sitka Sound (southern site, red) (*a*), as well as sites sampled across the latter two time periods (2003/2009 versus 2018/2019, respectively) in each location (*b* and *c*). The maps only display sites from the latter two sampling periods. (Online version in colour.)

local base of operation by small boat (roughly 10–18 km). The locations of the sites surveyed in 1988 in Sitka Sound were recorded by hand on a navigational chart, which was not used in the second resampling effort and could not be located prior to the third sampling effort. In the second set of sampling periods (2003 for Torch Bay; 2009 for Sitka Sound), sites were located in the same manner from the same areas as in the 1988 surveys, but were assigned high-resolution latitude and longitude coordinates using GPS. The 2018 data from Sitka Sound and 2019 data from Torch Bay were obtained from these same GPS locations (*n* = 11 for Torch Bay; *n* = 16 for Sitka Sound). The hand-drawn map of Sitka Sound from 1988 was relocated in 2021, and the locations were extracted by hand using Google Earth.

Sea urchins and macroalgae were sampled in 0.25 m² quadrats, placed randomly on the seafloor along the 6–7 m isobath. We sampled approximately 20 such quadrats for macroalgae and sea urchins at each site. The test diameters of sea urchins were measured until greater than 200 individuals or 20 quadrats had been sampled. From these measurements, we determined the density and size–frequency distribution of sea urchin populations and the density of kelp species (including *Macrocystis pyrifera*, *Nereocystis leutkeana*, *Pleurophycus gardneri*, *Agarum clathratum*, *Neoagarum fimbriatum* and the category *Laminaria* spp.), as well as community structure. Community structure was sampled in the same quadrats used to count kelps by estimating the percentage cover of primary benthic space holders, which were primarily coralline algae and other fleshy macroalgae, including kelps. Each taxon was given a score of 1–6, which represented (i) less than 5% cover, (ii) 5–25% cover, (iii) 26–50% cover, (iv) 51–75% cover, (v) 76–95% cover or (vi) greater than 95% cover [4]. Because we were interested in understanding the effect of sea urchin grazing on the algal assemblage, we estimated

sea urchin biomass density for all sampled site × year combinations. This is especially important because sea urchin density, another potential estimator of grazing pressure, is based on abundance, which is likely to be inadequate for estimating the effect of urchins on the algal community when the size distributions of both urchin species were as broad as found in this investigation (electronic supplementary material, figure S1). To transform numerical density to biomass density, we first estimated volume for both red (*Mesocentrotus franciscanus*) and green (*Strongylocentrotus droebachiensis*) urchins using the equation for a hemisphere (equation (2.1)), then converted volume to mass using the 1 to 1 relationship between wet mass (g) and volume (cm³) that has been previously described (see mass/diameter equation in [4]), and finally converted these values to biomass density (kg m⁻²):

$$\sum_{i=1}^{n} \frac{2}{3}\left(\frac{D}{2}\right)^3 \pi \left(\frac{1\,\text{liter}}{1000\,\text{cm}^3}\right)\frac{1\,\text{kg}/1\,\text{liter}}{A}, \quad (2.1)$$

where $D$ = test diameter (cm), $A$ = area sampled (m²) and $n$ = number of urchins sampled.

## (d) Statistical analyses

To compare both sea urchin biomass density and kelp density across years, we used an ANOVA with year as a fixed effect. For these analyses, the mean values for kelp density or urchin biomass density were calculated over all quadrats sampled at a site, and those averaged values were used in subsequent analyses. When *year* was significant, we used a Tukey HSD test to determine differences among specific years. Data were log-transformed (log [$x$ + 1]) as needed to meet the assumptions of normality and homogeneity of variances. We also present size–frequency

distributions for urchins to assess the degree to which any observed difference in biomass density of urchins over time was caused by population density versus size distribution. To determine if the composition of the algal assemblage changed over time, we used a PERMANOVA analysis, and to assess spatial variability in assemblage structure within locations (e.g. patchiness), we used a PERMDISP analysis. For these multivariate analyses, we used location × year combinations as levels of a single factor ($n = 6$ levels). To assess the potential effects of otter harvests on local otter abundances and the subtidal community, we used regression approaches to determine how sea otter sighting indices (see above), sea urchin biomass density and kelp density covaried with Euclidean distance from the town of Sitka. This last analysis of the otter sighting data was only done for the 2018 sampling period in Sitka Sound because otters were not harvested from Torch Bay and spatially explicit measurements of otter presence were not available for other areas and earlier years. Euclidean distance from a central point in Sitka was used because there are three harbours in Sitka Sound and numerous islands that provide several different estimates of potential distance hunters could travel on the water to a survey site. Using this Euclidean distance analysis, we discovered significant distance relationships for otters, urchins and kelp in 2018 and thus conducted the similar distance analyses for urchins and kelp for the 2009 surveys (prior to any significant sea otter harvests) in Sitka Sound. Finally, we directly compared the otter sighting indices with total urchin biomass density and kelp density for 2018 in Sitka Sound. For regression analyses, we fit both linear and non-linear (square root transform) models and compared the model fits using $R^2$-values (electronic supplementary material, table S1). Here, one-tailed tests were used because each comparison had a directional hypothesis (e.g. a negative relationship between urchin biomass density and otter presence).

Statistical analyses and tests (critical $\alpha = 0.05$) were run in JMP Pro 14 (v. 14.0.00) or PRIMER-E (v. 7) for community analyses.

# 3. Results

## (a) Sea otters

Southeast Alaska supported an estimated 5407 (4053–6855, 95% CI) sea otters in 1988 [22], and in Sitka Sound, there were low numbers of animals mostly limited to the north and south peripheries of the outer Sound [28]. Otters increased in abundance through the 1990s (judging from modelled projections [22] and reports by local residents) and by 1995 the population for all of southeast Alaska contained an estimated 8027 (5578–10 751, 95% CI) animals. By the time of our first resampling in 2009, Southeast Alaska contained an estimated 22,271 (16 749–28 544, 95% CI) sea otters, 639 (311–1125, 95% CI) of which occurred in Sitka Sound (area N05 in fig. 1 of [22]). Otters were commonly observed in the nearby waters of our Sitka Sound sites during resampling activities in 2009. The Torch Bay area (N01 from [22]) supported an estimated 160 otters in 1988, a number that has remained roughly constant to present (see electronic supplementary material, figure S2 for relative densities through time).

Two thousand, seven hundred and forty-four sea otters were harvested from the Sitka Sound area (N05 from [22]) between 1989 and 2015 [23]. Harvest numbers increased from 53 yr$^{-1}$ from 1989–2009 to 272 yr$^{-1}$ from 2010–2015. This increasing harvest mortality caused the local population to decline from approximately 900 animals in the early 2000s to less than 500 animals by 2012 (the final year of Tinker

et al.'s [22] analysis). Without this harvest, the Sitka Sound sea otter population is projected to have increased to over 1300 animals by 2012 [23]. These analyses thus indicate that just prior to our 2018 sampling, the Sitka Sound sea otter population density was approximately 70% lower than it would have been in the absence of harvest. Our 2018 surveys establish that the likelihood of sighting an otter increased with distance from the town of Sitka, with a 300% increase in the probability of seeing an otter at the sites most distant from Sitka compared to those closest to town (figure 2; $t_{12} = 4.10$, $p < 0.001$).

## (b) Sea urchins
### (i) Sitka Sound
In 1988, red and green urchins were large and abundant ($\bar{x} = 1.5$ kg m$^{-2}$, s.e.m. = 0.2) in Sitka Sound (figure 3; electronic supplementary material, figure S1). Although green urchin abundance had increased somewhat by 2009, red urchins were essentially absent from Sitka Sound at this time. While there was a detectable decrease in urchin biomass density with increasing distance from Sitka (figure 2; $F = 13.53$, d.f. = 1, 14; $p = 0.0013$), total urchin biomass density had declined by 99% from 1988 across the area. By 2018, total urchin biomass density had increased to 0.25 kg m$^{-2}$ (s.e.m. = 0.083; means = 0.164 and 0.086 kg m$^{-2}$ for red and green urchins, respectively; figure 3), with total urchin biomass density showing a decrease with increasing distance from town (figure 2; $F = 5.16$; d.f. = 1, 14; $p = 0.0197$). When compared directly, we found a decline in biomass density of urchins with the increasing probability of sighting an otter (figure 4; $F = 8.23$; d.f. = 1, 14; $p = 0.006$), with a decline in urchin biomass density occurring when the probability of seeing an otter surpassed 0.5.

### (ii) Torch Bay
In 1988, red and green urchins were small and rare in Torch Bay (0.0005 kg m$^{-2}$), and in 2003, these patterns were largely unchanged (figure 3; electronic supplementary material, figure S1). In 2019, the size structure of urchins remained largely unchanged (electronic supplementary material, figure S1), whereas sea urchin biomass density increased almost 200% to 0.0014 kg m$^{-2}$ (figure 3).

## (c) Macroalgae
### (i) Sitka Sound
Kelps were essentially absent from all sites in 1988 (figure 3). By 2009, however, kelp density had increased to 21 individuals per m$^2$ (s.e.m. = 2.45). One or more individuals occurred in most of the quadrats sampled, and at this time, there was no significant pattern of variation in kelp density with distance from the town of Sitka (figure 2; $F = 1.96$, d.f. = 1, 14; $p = 0.092$). By 2018, total kelp abundance had declined about 60% from 2009 to 8 individuals m$^{-2}$ (s.e.m. = 2.27), and one or more individuals occurred in less than half of the quadrats sampled. Moreover, kelp density increased with increasing distance from the town of Sitka (figure 2; $F = 3.51$; d.f. = 1, 14; $p = 0.042$), as well as with the probability of seeing an otter (figure 4; $F = 6.43$, d.f. = 1, 14; $p = 0.012$). Reflecting the pattern observed in urchin biomass density, kelp density was consistently low at sites where the probability of seeing an otter was less than 0.5.

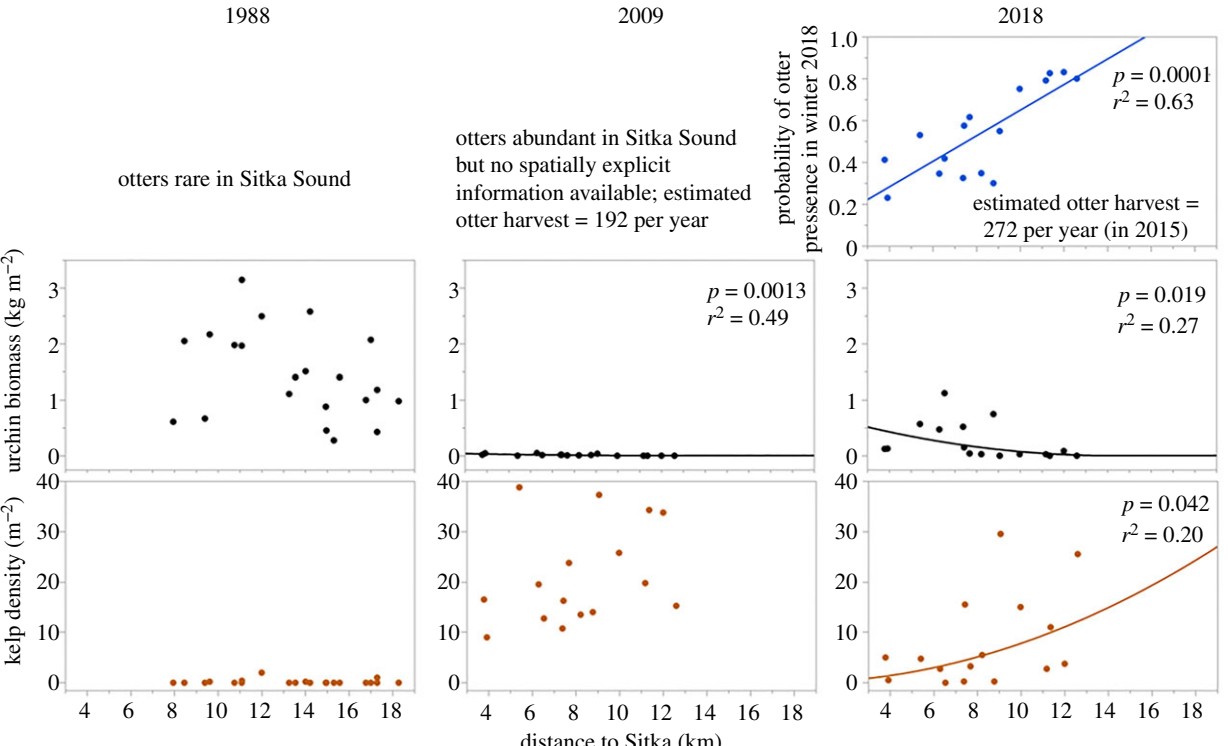

**Figure 2.** Patterns of variation with distance over time from the town of Sitka for sea otter abundance index, urchin biomass density and kelp density. The line indicates a significant linear (2018 otters) or nonlinear relationship with distance from Sitka. With the exception of the relationship between the distance to the town of Sitka and the probability of otter presence, all analyses shown here were performed on square-transformed data. (Online version in colour.)

### (ii) Torch Bay

Kelps were abundant (33 individuals m$^{-2}$; s.e.m. = 7.8; figure 3) throughout Torch Bay in 1988. One or more individuals occurred in 72% of the quadrats sampled at this time. Many individuals were small. Kelp density had declined about 35% to 21.5 individuals m$^{-2}$ by 2003 (s.e.m. = 4.1; figure 3), at which time one or more individuals occurred in 89% of the quadrats sampled. Kelp density had declined further by 2019 to 12.1 individuals m$^{-2}$ (s.e.m. = 2.4), at which time one or more individuals occurred in 82% of the quadrats sampled.

### (d) Ecosystem state

Both sea urchin and kelp abundance varied greatly in time and space over the 30-year time series of measurements. In general, the relationship between sea urchin density and kelp density resulted in the community being defined by two distinct areas of state space, one in which urchin biomass density is uniformly low and kelp density is high but variable (referred to hereafter as the kelp state), and the other in which kelp density is uniformly low and urchin biomass density is high but variable (referred to hereafter as the urchin state; figure 5).

### (i) Sitka Sound

In 1988, mean urchin biomass density was greater than 1 kg m$^{-2}$ at over 60% of the sites. All sites were in the urchin state at this time. By 2009, mean urchin biomass density was less than 0.05 kg m$^{-2}$ at all sites (nearing 0 kg m$^{-2}$ at most of these), resulting in all sites being in the kelp state. However, by 2018, mean urchin biomass density varied between 0 and 1 kg m$^{-2}$ across the sites, resulting in about half of these sites being in the kelp state and the other half

nearing the point of transition between the two states (i.e. both urchin biomass density and kelp density were relatively low) or had moved slightly into the urchin state (i.e. urchin biomass density was distinctly elevated and kelp density was low).

### (ii) Torch Bay

All Torch Bay sites remained distinctly within the kelp state throughout the 31-year time series (i.e. 1988–2019; figure 5).

### (e) Community structure

Benthic community structure varied substantially over space and time (PERMANOVA pseudo $F_{5,107}$ = 27.49, $p$ < 0.001; figure 6), with all pairwise comparisons of location × year differing significantly from each other ($p$ = 0.001). The greatest of these pairwise differences was between Sitka Sound and Torch Bay in 1988. These results are illustrated in the non-metric multidimensional scaling plots, where distance in two-dimensional space indicates differences in the community structure. This plot also demonstrates the significant differences in variability in community structure within locations (Sitka Sound versus Torch Bay) in each year of sampling (PERMADISP $F_{5,107}$ = 13.641, $p$ = 0.001; figure 6), which is illustrated by the size of the ellipse. Temporal differences within Sitka Sound versus Torch Bay were most evident from this analysis. For Sitka Sound, the spatial variability in community structure was similarly low between 1988 (urchin-dominated) and 2009 (algal-dominated) ($p$ = 0.122), but different ($p$ < 0.0010) and very high for 2018 (i.e. some sites urchin-dominated and some sites algal-dominated). For Torch Bay, the variability of the community did not change over time (all pairwise comparisons $p$ > 0.30) even though the composition of the communities changed—implying that

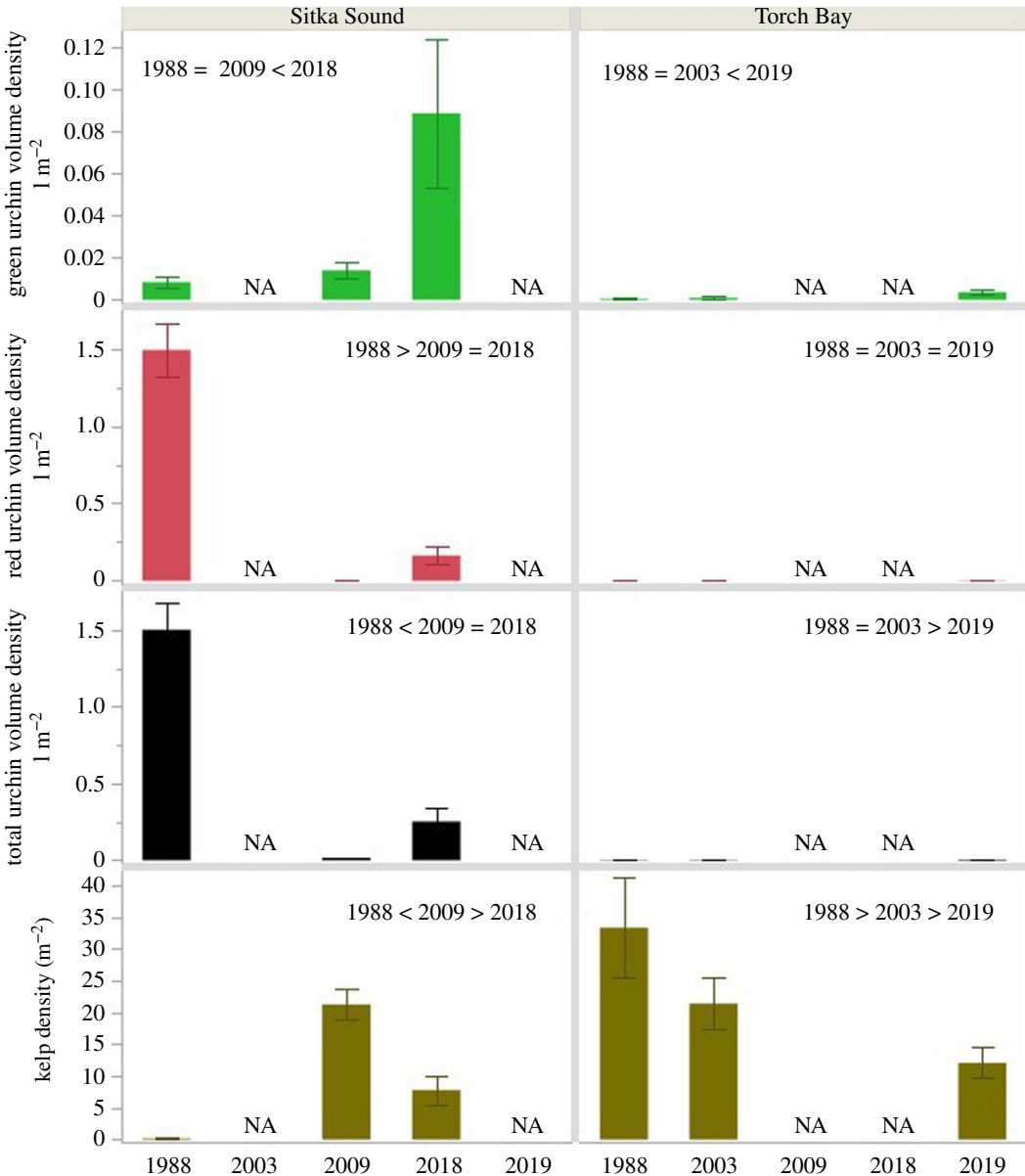

**Figure 3.** Variation in abundance of sea urchins and kelp in Sitka Sound and Torch Bay among the years. Green = *Strongylocentrotus droebachiensis*, Red = *Mesocentrotus franciscanus*, Black = total urchin biomass density, Brown = *Laminariales*. Inset year comparison is from ANOVA/TUKEY analyses. (Online version in colour.)

whatever led to the change in community structure affected the entire location.

## 4. Discussion

Our understanding of the otter–urchin–kelp trophic cascade is a product of both theory [29,30] and data from spatio-temporal contrasts of habitats with and without sea otters or areas that differ in the timing of otter occupancy. Here, we use similar data gathered at multiple temporal and spatial scales to show how the local harvesting of sea otters appears to have mediated the outcome of this trophic cascade, resulting in within-region variability of kelp density and community structure in an area with a large sea otter population. Our data from Sitka Sound in 1988 (when otters were still recolonizing and at very low densities) and 2009 (after this area had been occupied by sea otters for several decades), while consistent with the well-known otter–urchin–kelp paradigm, are nonetheless remarkable because of their extreme difference [4]. However, the data from 2018

provide new insight into this trophic cascade through the influence of human harvest in Sitka Sound on the probability of seeing an otter. While it is clear that the ecosystem exists in one of two alternate states at the equilibria (otters absent and otters near carrying capacity), we document a wider range of community states within Sitka Sound in 2018, including some kelp dominated sites and some urchin-dominated sites. These findings highlight the potential for small-scale variation in the presence of sea otters to create patchiness in the kelp forest landscape that may allow for the co-management of kelp forests and shellfisheries in areas with otters.

Anecdotally, we understand that the sea otter harvest has been greatest closest to the town of Sitka, supported by previous analyses of otter harvests in Sitka Sound [23], which we hypothesize created a more spatially explicit pattern in community structure and ecosystem state than was otherwise expected. We found the sites with the fewest urchins were farther from the town of Sitka, whereas the sites with the least kelp were closest to the town—although there was some important variability in this relationship (electronic supplementary material, figure S3). In particular, we found

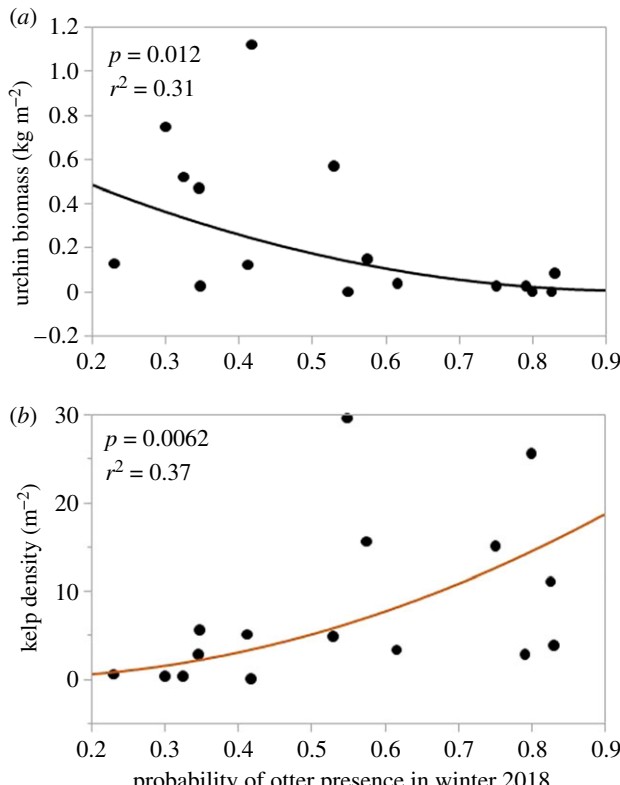

**Figure 4.** Functional relationships between urchin biomass density (*a*) and kelp density (*b*) as a function of the probability of seeing an otter in Sitka Sound in 2018. (Online version in colour.)

some intermediate states (with some urchins and low kelp density) scattered throughout the region. While it is unclear whether the intermediate states are at equilibria, or in the process of changing states, the overall patchiness in the density of urchins and kelp in 2018 indicates that both ecosystem states can co-occur when the presence or relative density of otters is patchy. These localized sea otter effects are consistent with a growing recognition that sea otter habitat use and abundance are often structured at very small spatial scales [22,31,32]. This spatial structuring occurs because reproductively mature sea otters, particularly females, have small lifetime home ranges of just 10–25 km of coast [31,33,34], with limited movements of reproductive individuals between adjacent habitats [35], and thus substantial differences in abundance can occur over short distances [32]. Sea otter responses to top-down threats (whether human harvesters, killer whales or white sharks) can also reflect small-scale variation in the risk landscape [36,37], whereby otters change their behaviours and habitat usage in response to threats. Understanding how spatially varying mortality risk for sea otters can translate into patchiness in community structure may help explain archaeological evidence that indigenous people in the Pacific Northwest apparently had access to areas of both abundant shellfish and abundant sea otters [10,11,38]. Given previous research on the alternative stable states associated with the sea otter–urchin–kelp trophic cascade, we suggest that management actions promoting patchiness in sea otter occupancy seem feasible and may be important for maintaining both kelp ecosystem services and shellfisheries in regions with abundant otters.

Although the primary focus of our study was on the ecological consequences of the recovery and subsequent reduction of sea otter populations in southeast Alaska, other processes no doubt contributed to the large-scale patterns of variation in the distribution and abundance of sea urchins and kelp that occurred over the course of our study. Of particular importance is the loss of sunflower stars (*Pycnopodia helianthoides*) because of sea star wasting disease (SSWD) and the episodic recruitment of sea urchins [39]. The extreme difference in urchin and kelp abundance between 1988 and 2009 in Sitka Sound, while mainly caused by the repatriation of sea otters into an area from which they had been absent for more than a century, may have been exacerbated by a lack of urchin recruitment (at least in the years immediately prior to 2009) and a robust population of sunflower stars that consumed most newly recruited small urchins that entered the system during the 1988 to 2009 period [40,41]. The more detailed time series required to chronicle these effects is lacking from our study locations in Sitka Sound and Torch Bay. However, D. O. Duggins never witnessed the recruitment of otherwise abundant red sea urchins during the 5 or 6 years he worked in Torch Bay in the late 1970s and early 1980s ([4] and personal communication), and we see no indication of a recruitment pulse in the size–frequency distribution of sea urchins from Sitka Sound in 1988 (electronic supplementary material, figure S1). Recent studies from other localities [40,42] suggest sunflower stars can affect the distribution and abundance of urchins and kelp, and it is possible that SSWD contributed to shifts in community structure seen across both Torch Bay and Sitka Sound.

Torch Bay provides an intriguing point of contrast with Sitka Sound because sea otters remained at or near carrying capacity in Torch Bay throughout the time series. Although urchin biomass density increased and foliose algae and kelp density declined somewhat in Torch Bay between 2003 and 2019, the system remained distinctly in the algal/kelp state throughout our three decades of study (figure 5*b*). And while urchin biomass density in Torch Bay increased, that increase did not approach the 0.5–1 kg m$^{-2}$ levels associated with the intermediate sites with some urchins and some kelp in Sitka Sound.

We hypothesize that the reduction in kelp density that occurred in Torch Bay between 1988 and 2003 was the likely result not of grazing, but of heavy kelp recruitment following the repatriation of otters to this area just before 1988 followed by succession to a mature kelp forest. The further reduction in kelp density and shift in community structure that occurred between 2003 and 2019 could be a consequence of continued succession [43] and/or the loss of the sunflower star from SSWD releasing some pressure on key kelp forest grazers, including both snails and small urchins [40]. However, it would be surprising if the increase in urchin biomass density that occurred in Torch Bay during this latter time period was an important contributing factor to the change in kelp density, given the overall low biomass density in sea urchins in comparison with Sitka Sound (figure 3). Indeed, because the community structure in Torch Bay in 2019 became more similar to the community structure in Sitka Sound in 2018 (figure 6), we hypothesize that the driver was probably something occurring region wide (e.g. SSWD or other anomalous environmental conditions such as the Blob [44]).

Because urchins and kelps were sampled independently from different quadrats, we cannot assess the pattern of covariation in urchin biomass density and kelp density at this smallest spatial scale. Nevertheless, we can assess the patterns of covariation in urchin and kelp abundance at the

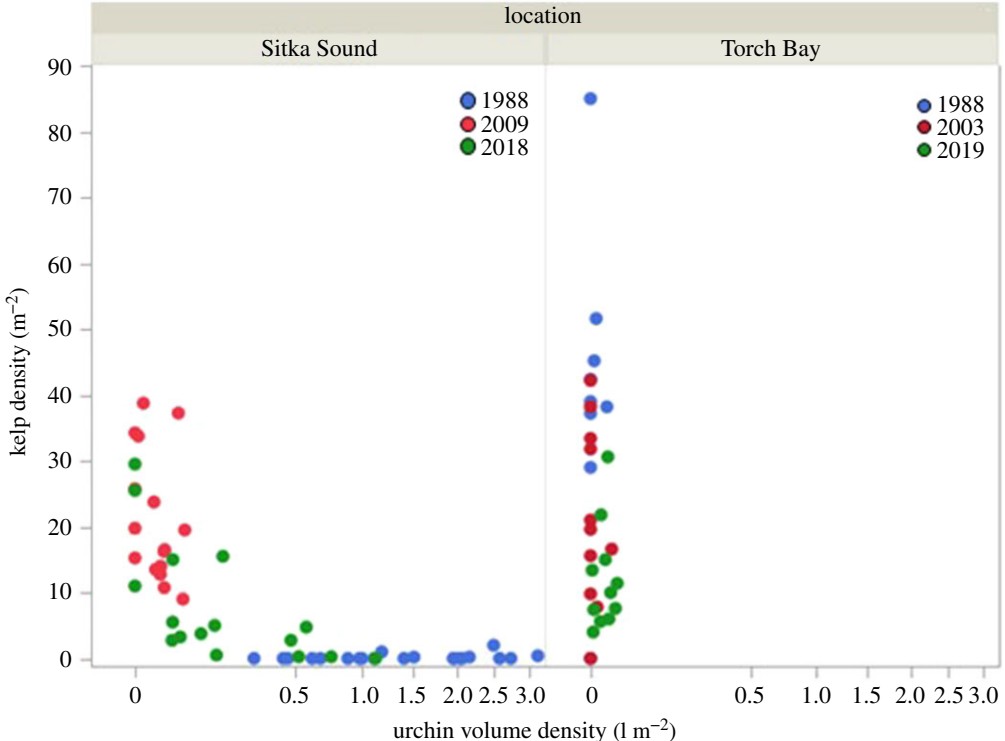

**Figure 5.** State space plots of total urchin biomass density versus kelp density by sample sites for Sitka Sound and Torch Bay among the years these areas were sampled. (Online version in colour.)

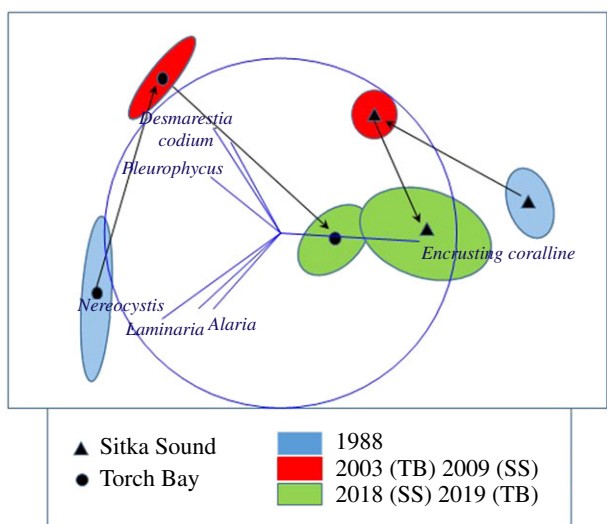

**Figure 6.** Non-metric multidimensional scaling plot of the centroids and 80% confidence ellipses for benthic community structure for Sitka Sound and Torch Bay based on the broad taxonomic and functional groups. Black arrows depict the temporal trajectory of the communities at each location (1988 to 2003 (TB) or 2009 (SS) to 2018). Blue vectors indicate the region of the graph with greater abundances of indicated species. (Online version in colour.)

scale of sites within our two study regions, and at that scale, the system generally occurs in one of two distinct alternate states (the urchin state or the kelp state; figure 5). Those sites in Sitka Sound with intermediate urchin and kelp densities in 2018 may have been in a state of transition, where a decrease in the abundance of otters near the town of Sitka led to a transition toward the urchin-dominated state. This conjecture is supported by the lack of sites exhibiting both high urchin biomass density and high kelp density (figure 5) and raises an important point for consideration if management actions were taken to promote patchiness in

otter occupancy. These findings lend further support to the view that North Pacific kelp forests occur as alternate stable states [10,19,45,46], with the transition points between these states being both rare and unstable [47]. This situation contrasts sharply with that for kelp forests in Australia and New Zealand where kelps and urchins typically co-occur at relatively high densities, even at small spatial scales [48].

Our study is founded on post hoc interpretations of simple time-series measurements that are informative because of the large spatial and temporal scales over which the information was obtained, and the interceding events (sea otter recovery initially, and subsequent sea otter reduction from harvests) that made the observed patterns of change interesting. Our analyses and interpretations lack the inferential rigour of well-designed and properly controlled experiments. However, experimental studies of processes that occur at such large scales of space and time were simply not possible in this case and, by analogy, will not be possible in many others in which the scales of process are similar. Progress in field ecology demands recognition of the fact that, from a methodological perspective, one shoe does not fit all.

Data accessibility. The data that support the findings of this study are openly available in BCO-DMO at www.bco-dmo.org, project number 756735.

Authors' contributions. T.R.G.: conceptualization, data curation, investigation, methodology, writing—original draft and writing—review and editing; S.C.R.G.: conceptualization, investigation, methodology and writing—review and editing; M.R.L.: conceptualization, investigation, methodology, writing—review and editing; U.H.: investigation and supervision; J.A.E.: conceptualization, data curation, funding acquisition, methodology, writing—original draft and writing—review and editing; P.T.R.: conceptualization, data curation, formal analysis, investigation, methodology, supervision, visualization and writing—review and editing; M.T.T.: formal analysis, visualization and writing—review and editing; M.C.K.: data curation, investigation, methodology and writing—review and editing; K.J.K.: conceptualization, funding

acquisition, investigation, methodology, project administration, supervision, writing—original draft and writing—review and editing. All authors gave final approval for publication and agreed to be held accountable for the work performed therein.

Competing interests. We declare we have no competing interests.

Funding. This work was supported by the David and Lucile Packard Foundation (K.J.K. and P.T.R.), UC Santa Cruz (K.J.K. and P.T.R.), The Alfred P. Sloan Foundation (K.J.K.), The National Science Foundation (OCE-1752600), the National Park Service and US Geological Survey.

Acknowledgements. We thank J. Bodkin, D. Carney, G. Esslinger, K. Miles, D. Monson, J. A. Toy, E. O'Brien, P. Tate and G. VanBlaricom for their survey efforts and recent contributions to this study; M. Kisslinger, R. B. Benter and C. Putnam at US Fish and Wildlife Services; D. O. Duggins, J. Tomoleoni, S. Lummis, W. Raymond and N. Laroche for their knowledge, observations and resources that improved the quality of this manuscript. We would also like to thank Sitka Sound Science Center and C. Gray for dive operations; S. R. Clabuesch, T. White, E. M. Donham and L. Strope for additional field assistance with this project. We dedicate this paper to Dr Umihiko Hoshijima, who tragically lost his life in a diving accident while participating in this study. Umi's friendship, joyful nature and commitment to research and mentoring students continue to inspire us.

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
