## [Peer Review File · Proceedings of the Royal Society B: Biological Sciences]

Review History

RSPB-2020-3001.R0 (Original submission)

Review form: Reviewer 1

Recommendation

Major revision is needed (please make suggestions in comments)

Scientific importance: Is the manuscript an original and important contribution to its field?

Acceptable

General interest: Is the paper of sufficient general interest?

Excellent

Quality of the paper: Is the overall quality of the paper suitable?

Acceptable

Is the length of the paper justified?

Yes

Should the paper be seen by a specialist statistical reviewer?

No

Do you have any concerns about statistical analyses in this paper? If so, please specify them explicitly in your report.

No

It is a condition of publication that authors make their supporting data, code and materials available - either as supplementary material or hosted in an external repository. Please rate, if applicable, the supporting data on the following criteria.

Is it accessible?

No

Is it clear?

N/A

Is it adequate?

No

Do you have any ethical concerns with this paper?

No

Comments to the Author

This manuscript describes spatial and temporal patterns in kelp and urchin biomass in SE Alaska, and relate them to patterns in otter abundance. They show patterns consistent with the classic kelp-urchin-otter trophic cascade and suggest the ecosystem implications otter harvesting, even without extirpation. They found that in Sitka Sound (an area that became recolonized by an abundance of sea otters) patterns in otter presence over time, and over space in the most recent survey period, is positively related to kelp density and negatively related to urchin biomass. Additionally, they found that the algal community in Sitka Sound became more spatially heterogeneous over time. In Torch Bay, where otter abundance is low, they found high kelp biomass in all time periods, an increase in small urchins in the latest time period, and a changing algal community over time with no significant change in spatial community dissimilarity. The authors go on to explain these patterns via trophic cascade, but also make an attempt to relate them to the loss of Pycnopodia from Sea Star Wasting Disease.

Strong ecological baseline data such as this is valuable and rare. The role of trophic redundancy in shaping ecosystems is important from both a theory and management standpoint. However, without data on Pycnopodia over time, one cannot rule out alternate drivers, such as oceanographic changes from the Warm Blob. Per Konar et al 2019 and Harvell et al 2019, Pycno only became absent after SSWD in some areas, and could still persist as a significant part of the community. Reference to data on sea star abundance in the study area would be vital to bolster the related claims. The SSWD story is likely important here, but unfortunately lacks data. However, the authors present interesting data that suggests the ecosystem effects of otter harvest. In my opinion, the manuscript would be strengthened by a much stronger focus on that aspect of the story in the abstract, intro, and discussion. The resultant paper would be important from a management standpoint.

Additional comments:

Otter encounters increased with distance from Sitka, which the author's attribute to harvest patterns. Could this be related to other patterns (oceanographic, vessel traffic, etc)? Some background information of site fidelity in otters would also be helpful. Could the pattern of increasing otter encounter at more kelp-y locations be due to otter habitat preference?

Figs 2, 4 and associated results: what are the goodness of fit values for the regressions? These would help evaluate the effect of distance from town.

Fig 3: an added panel of otter population over time would be useful here. Additionally, include indication where there is no data, so as to not confuse with the absence of one of the groups during the time period.

Review form: Reviewer 2

Recommendation

Reject – article is not of sufficient interest (we will consider a transfer to another journal)

Scientific importance: Is the manuscript an original and important contribution to its field?

Good

General interest: Is the paper of sufficient general interest?

Acceptable

Quality of the paper: Is the overall quality of the paper suitable?

Good

Is the length of the paper justified?

Yes

Should the paper be seen by a specialist statistical reviewer?

Yes

Do you have any concerns about statistical analyses in this paper? If so, please specify them explicitly in your report.

Yes

It is a condition of publication that authors make their supporting data, code and materials available - either as supplementary material or hosted in an external repository. Please rate, if applicable, the supporting data on the following criteria.

Is it accessible?

N/A

Is it clear?

N/A

Is it adequate?

N/A

Do you have any ethical concerns with this paper?

No

Comments to the Author

General comments : This manuscript addresses interesting questions on the spatial and temporal dynamics of a keystone predator, building on a well-studied system, namely the sea otter-urchin-kelp trophic cascade in Alaska. The study spans an unusually long period of time (several decades), and although the spatial extent is less noteworthy (two areas), there is an interesting contrast between them and the sampling within each area is impressive considering the effort needed for underwater research, especially in remote, cold waters.

Unfortunately, the main premise – an assessment of the relative importance of an alternative keystone predator, a sea star – is not supported by any data, and it is simply assumed that any impact of this mesopredator has been removed by the pandemic disease that has affected many sea star species in the northeast Pacific. While the use of such a natural experiment is not in itself a flaw, especially given the temporal scale of the study, there are limitations in the interpretation of the patterns observed. In particular, a comparison of just two areas is always a concern (i.e. no replication in the statistical sense) as they can differ in many ways. Likewise, temporal changes are difficult to interpret when they occur over the entire system of interest. This is not to say that I am not convinced that this study provides evidence for a minor role of the sea star, but I see this aspect more as a discussion point rather than as the main thrust of the manuscript. Also, given the inherent noise in natural systems, I am not certain how strong a sea-star signal would need to be detectable.

I am also not convinced that the relationships are sufficiently ‘non-linear’ to be described as a tipping point (see specific comments below).

Thus, overall, I found the work to add another chapter to a fascinating and well-developed story, but perhaps without remarkable novelty (I actually found the gradient of hunting pressure to be the more interesting aspect of the study). To attract a wider audience, I think the authors would need to balance the manuscript better, and if they do not have adequate data of their own, perhaps incorporate those from earlier studies to better justify the intriguing question of the relative importance of different predators.

Specific comments:

Questions (line 35): references are needed to support this assertion, i.e., who has been asking these questions?

Autotrophic kelp (line 48): all kelp are autotrophic, thus this adjective not needed.

Sea star data (lines 77 & 78): this is all that is offered in terms of sea star population changes. The references are either older (21) or about the disease (25) and the rest is unpublished or observational. Again, fine to support a discussion point, but not for “disentangling” (line 14) the relative effects of these two predators.

Sea otter densities: Similarly, the data for sea otter is limited – either all or none, an index taken at a single moment in time (Figure 2) or assumed to be constant (Torch Bay). Moreover, it is hard to compare the two areas – data are provided at a population level and not as true densities, e.g., otter/km² or km of shoreline (lines 172+).

Hunter behavior (line 87): how is this known? Perhaps it is more of a reasonable assumption, but if that is the case, it should be stated as such.

Statistics: I am not an expert, but I think that considering Year to be a fixed factor (line 143) needs justification (usually a random factor). Moreover, treating location-year combinations as levels within a single factor (line 152) also seems unusual – typically a crossed design as least with traditional ANOVAs. This also needs support.

Background: the information in lines 171-190 do not seem to be the results of this study and would be better presented in the Introduction or Discussion (or perhaps Methods).

Distance (line 191): how was distance measured? Shortest water route, as the crow flies, or following the coast? Moreover, from Figure 1, there are clearly three close sites (all approximately 2-3 km from Sitka as the crow flies) whereas the graphs in Figure 2 show only two close sites but both are nearly 4 km away.

Urchin abundance (line 203): this trend seems “similar” only in being negative. The difference between the years is much more striking.

Sharp decline (line 206): I think that this is in the eye of beholder. There is certainly more scatter below (A) and above (B) the 0.5 value, but the variance is expected to increase as the mean does.

Decreases (e.g., line 230): I find here and elsewhere the use of percentages and folds to describe decreases nonintuitive. Instead of saying there is a “1.5 fold decrease”, I suggest

something along the lines of “decrease to 66% of previous levels”

State space (lines 235+): I think that the idea of alternative stable states within this system is well established and the non-linearity here is simply driven by the variation at the endpoints.

Variability in community structure (line 260): I am not certain what supports the statement that there the significant differences within locations – the size of the ellipses or their separation. Indeed, I find this figure to be of limited use, but that may be due being less familiar with multivariate analyses. Regardless, from this figure, it looks like Torch Bay has changed much more than Sitka Sound over time, which is not the message I have gotten from the other data and analyses. Perhaps this is being largely driven by changes in the species composition of the kelp assemblage, which is never discussed other than the mention on line 267.

Duggin’s observations (line 305): from what time period were these observations?

Combined effects/alternative explanations (lines 309-311): while I agree with this statement, I find it inconsistent with the claim that this study has “disentangled” the relative contributions of these three factors. Indeed, alternate explanations are numerous (lines 337-340 / lines 345-46), and again, the impact of sea stars is not resolved here as these alternatives can also explain the patterns observed. Indeed, the lack of kelp in Sitka Sound in 1988 is a strong suggestion that the sea star is not a keystone predator – the remaining observations in later years only reinforce this idea (i.e., the otter drives the show). The only evidence that I see otherwise is the 3x increase in abundance in Torch Bay from 2003 to 2018, but many other things could have changed over that 15-year period, and such changes are not uncommon just due to natural variation.

Coralline algae (Figure 6): these appear in this figure but are not mentioned elsewhere, in particular the methods – how were these quantified? If not density (difficult with encrusting organisms), then how can they be compared to the kelp? Likewise, *Codium* appears only in this figure.

Minor comments

Biomass vs. biomass density: the justification for using estimates of biomass is fine, but the term “biomass density” is confusion and not consistently used throughout the manuscript (sometimes “biomass” sometime just “density” (e.g, line 201).

Decision letter (RSPB-2020-3001.R0)

23-Feb-2021

Dear Dr Kroeker:

I am writing to inform you that your manuscript RSPB-2020-3001 entitled "Southeast Alaskan kelp forests: inferences of process from large-scale patterns of variation in space and time" has, in its current form, been rejected for publication in Proceedings B.

This action has been taken on the advice of referees, who have recommended that substantial revisions are necessary. The central issue has to do with a number of variables that are not adequately controlled for, specifically the lack of data for sea stars and the effects of hunting on sea otters. The referees are mixed in their possible suggestions as to how these issues could be dealt with. With this in mind we would be willing to consider a resubmission, provided the comments of the referees are fully addressed. However please note that this is not a provisional acceptance.

Sincerely,
Dr Daniel Costa
mailto: proceedingsb@royalsociety.org

Associate Editor
Board Member: 1
Comments to Author:

We received two constructive reviews, both of which commend the study for the long-term scope of the data, particularly underwater data from cold marine habitat. I agree with those points, but both referees also highlighted that the scope and conclusions are a major over reach with regards to the role of sea stars in the system. The fact that there are no actual data or analysis of sea stars, that the seastar wasting disease event was synchronous across areas, and any spatial contrasts are limited between just two areas, puts substantial limitations on what can be inferred about sea stars. In particular there is no spatial 'control' or contrast area to compare the SSWD event relative to any other broad spatial scale change. Such is the limitation of analyzing data from a 'natural experiment', which is not a criticism of the fieldwork effort, but rather a reality that confounding variables cannot be accounted for via the temporal and spatial contrasts available from the data. I agree with the referees that these weaknesses make components of the paper about sea stars a major over reach of what can be supported by the data and analyses, and that at most, the sea star part of the story is a minor discussion point. With that in mind, the results remain interesting but represent a minor incremental and confirmatory contribution to an already

well understood otter-urchin-kelp cascade. I'll also highlight the issue of confounding terminology of 'nonlinear' and tipping points that in this paper unfortunately perpetuates misuse and confusion. Yes it is true that some of the regressions are not straight lines, but that is not the same thing as nonlinear dynamics and the tipping points that can emerge due to bifurcations and unstable equilibria.

Reviewer(s)' Comments to Author:

Referee: 1

Comments to the Author(s)

This manuscript describes spatial and temporal patterns in kelp and urchin biomass in SE Alaska, and relate them to patterns in otter abundance. They show patterns consistent with the classic kelp-urchin-otter trophic cascade and suggest the ecosystem implications otter harvesting, even without extirpation. They found that in Sitka Sound (an area that became recolonized by an abundance of sea otters) patterns in otter presence over time, and over space in the most recent survey period, is positively related to kelp density and negatively related to urchin biomass. Additionally, they found that the algal community in Sitka Sound became more spatially heterogeneous over time. In Torch Bay, where otter abundance is low, they found high kelp biomass in all time periods, an increase in small urchins in the latest time period, and a changing algal community over time with no significant change in spatial community dissimilarity. The authors go on to explain these patterns via trophic cascade, but also make an attempt to relate them to the loss of Pycnospodia from Sea Star Wasting Disease.

Strong ecological baseline data such as this is valuable and rare. The role of trophic redundancy in shaping ecosystems is important from both a theory and management standpoint. However, without data on Pycnospodia over time, one cannot rule out alternate drivers, such as oceanographic changes from the Warm Blob. Per Konar et al 2019 and Harvell et al 2019, Pycno only became absent after SSWD in some areas, and could still persist as a significant part of the community. Reference to data on sea star abundance in the study area would be vital to bolster the related claims. The SSWD story is likely important here, but unfortunately lacks data. However, the authors present interesting data that suggests the ecosystem effects of otter harvest. In my opinion, the manuscript would be strengthened by a much stronger focus on that aspect of the story in the abstract, intro, and discussion. The resultant paper would be important from a management standpoint.

Additional comments:

Otter encounters increased with distance from Sitka, which the author's attribute to harvest patterns. Could this be related to other patterns (oceanographic, vessel traffic, etc)? Some background information of site fidelity in otters would also be helpful. Could the pattern of increasing otter encounter at more kelp-y locations be due to otter habitat preference?

Figs 2, 4 and associated results: what are the goodness of fit values for the regressions? These would help evaluate the effect of distance from town.

Fig 3: an added panel of otter population over time would be useful here. Additionally, include indication where there is no data, so as to not confuse with the absence of one of the groups during the time period.

Referee: 2

Comments to the Author(s)

General comments : This manuscript addresses interesting questions on the spatial and temporal dynamics of a keystone predator, building on a well-studied system, namely the sea otter-urchin-kelp trophic cascade in Alaska. The study spans an unusually long period of time (several decades), and although the spatial extent is less noteworthy (two areas), there is an interesting

contrast between them and the sampling within each area is impressive considering the effort needed for underwater research, especially in remote, cold waters.

Unfortunately, the main premise – an assessment of the relative importance of an alternative keystone predator, a sea star – is not supported by any data, and it is simply assumed that any impact of this mesopredator has been removed by the pandemic disease that has affected many sea star species in the northeast Pacific. While the use of such a natural experiment is not in itself a flaw, especially given the temporal scale of the study, there are limitations in the interpretation of the patterns observed. In particular, a comparison of just two areas is always a concern (i.e. no replication in the statistical sense) as they can differ in many ways. Likewise, temporal changes are difficult to interpret when they occur over the entire system of interest. This is not to say that I am not convinced that this study provides evidence for a minor role of the sea star, but I see this aspect more as a discussion point rather than as the main thrust of the manuscript. Also, given the inherent noise in natural systems, I am not certain how strong a sea-star signal would need to be detectable.

I am also not convinced that the relationships are sufficiently ‘non-linear’ to be described as a tipping point (see specific comments below).

Thus, overall, I found the work to add another chapter to a fascinating and well-developed story, but perhaps without remarkable novelty (I actually found the gradient of hunting pressure to be the more interesting aspect of the study). To attract a wider audience, I think the authors would need to balance the manuscript better, and if they do not have adequate data of their own, perhaps incorporate those from earlier studies to better justify the intriguing question of the relative importance of different predators.

Specific comments:

Questions (line 35): references are needed to support this assertion, i.e., who has been asking these questions?

Autotrophic kelp (line 48): all kelp are autotrophic, thus this adjective not needed.

Sea star data (lines 77 & 78): this is all that is offered in terms of sea star population changes. The references are either older (21) or about the disease (25) and the rest is unpublished or observational. Again, fine to support a discussion point, but not for “disentangling” (line 14) the relative effects of these two predators.

Sea otter densities: Similarly, the data for sea otter is limited – either all or none, an index taken at a single moment in time (Figure 2) or assumed to be constant (Torch Bay). Moreover, it is hard to compare the two areas – data are provided at a population level and not as true densities, e.g., otter/km² or km of shoreline (lines 172+).

Hunter behavior (line 87): how is this known? Perhaps it is more of a reasonable assumption, but if that is the case, it should be stated as such.

Statistics: I am not an expert, but I think that considering Year to be a fixed factor (line 143) needs justification (usually a random factor). Moreover, treating location-year combinations as levels within a single factor (line 152) also seems unusual – typically a crossed design as least with traditional ANOVAs. This also needs support.

Background: the information in lines 171-190 do not seem to be the results of this study and would be better presented in the Introduction or Discussion (or perhaps Methods).

Distance (line 191): how was distance measured? Shortest water route, as the crow flies, or following the coast? Moreover, from Figure 1, there are clearly three close sites (all

approximately 2-3 km from Sitka as the crow flies) whereas the graphs in Figure 2 show only two close sites but both are nearly 4 km away.

Urchin abundance (line 203): this trend seems “similar” only in being negative. The difference between the years is much more striking.

Sharp decline (line 206): I think that this is in the eye of beholder. There is certainly more scatter below (A) and above (B) the 0.5 value, but the variance is expected to increase as the mean does.

Decreases (e.g., line 230): I find here and elsewhere the use of percentages and folds to describe decreases nonintuitive. Instead of saying there is a “1.5 fold decrease”, I suggest something along the lines of “decrease to 66% of previous levels”

State space (lines 235+): I think that the idea of alternative stable states within this system is well established and the non-linearity here is simply driven by the variation at the endpoints.

Variability in community structure (line 260): I am not certain what supports the statement that there the significant differences within locations – the size of the ellipses or their separation. Indeed, I find this figure to be of limited use, but that may be due being less familiar with multivariate analyses. Regardless, from this figure, it looks like Torch Bay has changed much more than Sitka Sound over time, which is not the message I have gotten from the other data and analyses. Perhaps this is being largely driven by changes in the species composition of the kelp assemblage, which is never discussed other than the mention on line 267.

Duggin’s observations (line 305): from what time period were these observations?

Combined effects/alternative explanations (lines 309-311): while I agree with this statement, I find it inconsistent with the claim that this study has “disentangled” the relative contributions of these three factors. Indeed, alternate explanations are numerous (lines 337-340 / lines 345-46), and again, the impact of sea stars is not resolved here as these alternatives can also explain the patterns observed. Indeed, the lack of kelp in Sitka Sound in 1988 is a strong suggestion that the sea star is not a keystone predator – the remaining observations in later years only reinforce this idea (i.e., the otter drives the show). The only evidence that I see otherwise is the 3x increase in abundance in Torch Bay from 2003 to 2018, but many other things could have changed over that 15-year period, and such changes are not uncommon just due to natural variation.

Coralline algae (Figure 6): these appear in this figure but are not mentioned elsewhere, in particular the methods – how were these quantified? If not density (difficult with encrusting organisms), then how can they be compared to the kelp? Likewise, *Codium* appears only in this figure.

Minor comments

Biomass vs. biomass density: the justification for using estimates of biomass is fine, but the term “biomass density” is confusion and not consistently used throughout the manuscript (sometimes “biomass” sometime just “density” (e.g, line 201).

Author's Response to Decision Letter for (RSPB-2020-3001.R0)

See Appendix A.

RSPB-2021-1697.R0

Review form: Reviewer 1

Recommendation

Major revision is needed (please make suggestions in comments)

Scientific importance: Is the manuscript an original and important contribution to its field?

Good

General interest: Is the paper of sufficient general interest?

Good

Quality of the paper: Is the overall quality of the paper suitable?

Marginal

Is the length of the paper justified?

Yes

Should the paper be seen by a specialist statistical reviewer?

No

Do you have any concerns about statistical analyses in this paper? If so, please specify them explicitly in your report.

No

It is a condition of publication that authors make their supporting data, code and materials available - either as supplementary material or hosted in an external repository. Please rate, if applicable, the supporting data on the following criteria.

Is it accessible?

Yes

Is it clear?

Yes

Is it adequate?

Yes

Do you have any ethical concerns with this paper?

No

Comments to the Author

The manuscript deals with patterns of kelp, red urchin, and sea otters at two locations in Alaska over the past three decades – one where otters are hunted by humans, and one where they are not. The “classic” trophic cascade appears in the data, but with the added element of inferred hunting effects decreasing otter abundance closer to the town of Sitka. The research contributes to understanding the ecosystem dynamics associated with predator re-establishment and connections to human activities, which is an important management topic. The manuscript is well-written overall and most of the conclusions are well-supported, but it needs increased focus to form a cohesive narrative.

The impacts of sea otters on North Pacific kelp ecosystems is well-established. The authors should emphasize the new insight they bring to the table. The manuscript presents interesting data on the ecosystem effects of hunting sea otters - the main topic/question that the research

addresses. However, that thread is lost at many points in the narrative. This topic is of recent interest – see the recent Slade et al. publication*. I think this citation emboldens the authors' claims and should be incorporated into the manuscript introduction and discussion.

*Slade, E., McKechnie, I. & Salomon, A.K. Archaeological and Contemporary Evidence Indicates Low Sea Otter Prevalence on the Pacific Northwest Coast During the Late Holocene. *Ecosystems* (2021). <https://doi.org/10.1007/s10021-021-00671-3>

While the data provide increased understanding of the role of humans in shaping marine ecosystems, the implications of some of the results are overstated. There seems to be an underlying assertion that the manuscript deals with assessing otter population levels for management (lines 39-40, 305-314). This is a reach for the data presented and those sections should be re-written or removed.

The general edits suggested above and the specific comments below constitute a major revision.

Abstract

The first sentence does not capture the main thrust of the research and what makes it new and exciting - the influence of hunting.

L 19: The data presented do not a causal relationship

Background

The authors touch on many different topics in the background – trophic cascades, non-linear dynamics, and human-predator conflicts. While these are all interesting considerations in the story, the section needs work on guiding the reader towards a central question. This section would benefit from pronounced focus on how adding human impacts on otters (and adding humans to ecosystem conceptions in general) can contribute towards our understanding.

The connection made between urchins and sea otter-human conflicts is misleading as written (paragraph at L 42). Red urchins are not the only important recreational and commercial shellfish species who are otter prey. They are certainly not the most important per revenue or # of permits. A reader who doesn't know the system may think, based on the section written as-is, that the concerns over otters are all related to their consumption of urchins. Please clarify. Related to this – L58-60, I do not follow the logic to why this is the most important information need (compared to say the impact of otters on just shellfish). The research presented also includes kelp (an important foundation species, as mentioned) and hunting, so I suggest refocusing this paragraph. The authors mention the landscape of fear (and to the “risk landscape” in the Abstract), but this needs elaboration. While the data is not available to tease apart the direct vs indirect effects of hunting (numeric vs behavioral), the authors should do more to address the topic in both the background and discussion.

L 48-50: There are many ways that kelp benefit humans. Either add some more examples of ecosystem services or make the sentence more general.

L 53-54: “Limited” how? Spatially? Numerically? Please clarify.

L 56: Consider citing Carswell, L. P., S. G. Speckman, and V. A. Gill. 2015. Shellfish fishery conflicts and perceptions of sea otters in California and Alaska. Pages 333–368 in S. E. Larson, J. L. Bodkin, and G. R. VanBlaircom, editors. *Sea otter conservation*. Academic Press, Boston, Massachusetts, USA.

The last sentence of the section makes no mention of humans, and it should.

Methods

The methods appear sound and the authors do well pointing out the limitations of their dataset. If the approximate minimum distance of 1988 survey sites to Sitka is known, it should be included here. While the exact distributions of distances is unknown, any additional information on how 1988 survey sites were distributed with regards to distance from Sitka will be helpful for comparing to the later surveys.

Include if the boat was stationary/anchored during the otter surveys or if it was following a transect, grid, etc.

L 77: Citation needed for Torch Bay carrying capacity (as at line 93)

L 88-90: Is this inverse relationship from a citation or your own research?

Results

The results are thoroughly described, but some portions should be placed in the discussion. Some of the statements in this section are not solidly supported (see below).

“Non linear” (used in the discussion as well) doesn’t seem like the right phrase to describe some the results presented on ecosystem state. A threshold is apparent, but insistence on “nonlinear” is misleading. Perhaps “chaotic” or “variable”.

L 194-196: How did you calculate this number?

L 201: This should go in discussion, not results

Fig 2, panels for 2018 urchins and kelp - there are trends here, but low explanatory power. I see changes in variability over distance (which the authors rightly mention on L 251). L 318-319, 303-305 overstate the conclusions that can be drawn from this data.

Fig 3: Consider adding sea otter harvest numbers (L190-194) or density (Fig S2) to the Sitka side to really hammer the point home (the years won’t match up, but just create a new x axis).

L 213/ Fig 4: R-square needed. As with figure 2, there seems to be low explanatory power

L224-227: This sentence is confusing, partially from the phrase “indication of an increase”. I also do not see an “indication of an increase with distance” in Fig 2. Please re-write for clarity.

L237-238: Sentence should end at “small”, The last phrase is for the discussion

L 242-243: This should go in discussion, not results

Fig S3: include Sitka on map

Discussion

The authors clearly describe the ecosystem changes over time and space as a result of sea otter recolonization and sea otter hunting. They highlight the importance of spatial scale when looking at ecosystem patterns, which is important for management considerations. While it is well-written overall, more could be done to highlight the unique aspects of this study (as mentioned before), to tell a cohesive narrative, and there are a few conclusions that need more support.

L 293-294: Please remove or provide a citation. The present research doesn’t compare trophic cascades across systems.

L ~307-314: this section talks about how otters cannot coexist with productive shellfisheries, but the next paragraph is about how patchiness can allow for both to occur (and see Slade et al 201). Streamline between these two paragraphs. I re-emphasize that otter population thresholds and per-capita effects on shellfish are beyond the scope of this study – these aspects should be removed from the discussion.

L 352-354: This thought seems incomplete.

Review form: Reviewer 2

Recommendation

Reject – article is not of sufficient interest (we will consider a transfer to another journal)

Scientific importance: Is the manuscript an original and important contribution to its field?

Good

General interest: Is the paper of sufficient general interest?

Excellent

Quality of the paper: Is the overall quality of the paper suitable?

Good

Is the length of the paper justified?

Yes

Should the paper be seen by a specialist statistical reviewer?

Yes

Do you have any concerns about statistical analyses in this paper? If so, please specify them explicitly in your report.

Yes

It is a condition of publication that authors make their supporting data, code and materials available - either as supplementary material or hosted in an external repository. Please rate, if applicable, the supporting data on the following criteria.

Is it accessible?

N/A

Is it clear?

N/A

Is it adequate?

N/A

Do you have any ethical concerns with this paper?

No

Comments to the Author

General: I appreciate the efforts to revise the manuscript with respect to the sea star. However, although the "natural experiment" of a gradient in hunting pressure is a great addition to a well-established story, this aspect is as not well supported as others, which are more confirmatory of previous research, in spite of the enormous sampling efforts in the past and from another location. I feel that if it had been the focus, there would have been more effort here (e.g., sampling/data/local knowledge), and a more convincing case could have been made with less uncertainty with regards to nonlinearity and thresholds.

Specific comments

1. While I agree that there is overwhelming evidence for alternative community states (presented both here and in previous work) and that there must be some nonlinear relationships underlining it, I still do not see the data presented here being able to show a threshold value or suggest there are "threshold values in otter abundance" (lines 301 and 384). First, the otter abundance data are from one sampling (see additional comment and questions below) and were used to create "a spatial index" (line 104). These data are then shown as "probability of otter presence" in the figures, but how three successive counts of otter numbers are converted into probabilities is not given. Moreover, and more importantly, the link to the proposed threshold value of $K/2$ is not provided either (line 305). Perhaps I missed something here.

2. Otter sampling (lines 100-101): it is stated "We then repeated this sampling protocol two more times.", but it is not clear if this was done immediately (i.e., a single 5+5+5=15-minute sampling at each site) or on other days. If the latter, one would have more confidence in the abundance data. Indeed, more details are needed here (period - including dates - of sampling, how many days of sampling, time of day, weather, etc.) because if all the sampling was done over a short period, local conditions could influence the results.

3. Distance: I think the use of Euclidian distances is not ideal, and the shortest water route (passable by boats) is a more appropriate measure. Ideally, interviews with local hunters could

have provided this kind of information, but even navigation charts would be better. This is especially true if the study focusses on the relationship between distance and other variables. The harbour from which hunter depart should be indicated on the map.

4. Statistics: again, I am no expert, but with the emphasis on nonlinearities, then some better assessment is need to compare them with simple linear models, perhaps segmented/threshold regression or some goodness-of-fit.

5. Torch Bay changes: self-thinning is invoked to explain declines in kelp density, but even without knowing the species well, I doubt that their longevity spans decades.

6. Nonlinearity: The statement "functional relationships among otters, urchins, and kelp were nonlinear, resulting in alternate community states" in the Abstract is overreaching -- see previous comments and above.

Decision letter (RSPB-2021-1697.R0)

03-Sep-2021

Dear Dr Kroeker:

Your manuscript has now been peer reviewed. The reviews are quite mixed, while one reviewer thought the work was solid, he/she did not think that the research was sufficiently novel. In fact both reviewers didn't see what was this paper contributed to the sea otter kelp forest story. The other reviewer felt that the some of the statements were overstated and or the arguments could be better presented.

However, as someone who has worked with sea otters and watched the kelp forest of the west coast of the USA decline, I think your paper is important and timely. I am therefore providing you an additional opportunity to respond to the reviews. We do not normally allow multiple revision, but I am willing to give you one additional opportunity to respond to the reviewers comments.

Research ethics:

Use of animals and field studies:

It is a condition of publication that you make available the data and research materials supporting the results in the article (<https://royalsociety.org/journals/authors/author-guidelines/#data>). Datasets should be deposited in an appropriate publicly available repository and details of the associated accession number, link or DOI to the datasets must be included in the Data Accessibility section of the article (<https://royalsociety.org/journals/ethics-policies/data-sharing-mining/>). Reference(s) to datasets should also be included in the reference list of the article with DOIs (where available).

Please submit a copy of your revised paper within three weeks. If we do not hear from you within this time your manuscript will be rejected. If you are unable to meet this deadline please let us know as soon as possible, as we may be able to grant a short extension.

Best wishes,
Dr Daniel Costa
mailto: proceedingsb@royalsociety.org

Reviewer(s)' Comments to Author:

Referee: 1

Comments to the Author(s).

The manuscript deals with patterns of kelp, red urchin, and sea otters at two locations in Alaska over the past three decades – one where otters are hunted by humans, and one where they are not. The “classic” trophic cascade appears in the data, but with the added element of inferred hunting effects decreasing otter abundance closer to the town of Sitka. The research contributes to understanding the ecosystem dynamics associated with predator re-establishment and connections to human activities, which is an important management topic. The manuscript is well-written overall and most of the conclusions are well-supported, but it needs increased focus to form a cohesive narrative.

The impacts of sea otters on North Pacific kelp ecosystems is well-established. The authors should emphasize the new insight they bring to the table. The manuscript presents interesting data on the ecosystem effects of hunting sea otters - the main topic/question that the research addresses. However, that thread is lost at many points in the narrative. This topic is of recent interest – see the recent Slade et al. publication*. I think this citation emboldens the authors' claims and should be incorporated into the manuscript introduction and discussion.

*Slade, E., McKechnie, I. & Salomon, A.K. Archaeological and Contemporary Evidence Indicates Low Sea Otter Prevalence on the Pacific Northwest Coast During the Late Holocene. *Ecosystems* (2021). <https://doi.org/10.1007/s10021-021-00671-3>

While the data provide increased understanding of the role of humans in shaping marine ecosystems, the implications of some of the results are overstated. There seems to be an underlying assertion that the manuscript deals with assessing otter population levels for management (lines 39-40, 305-314). This is a reach for the data presented and those sections should be re-written or removed.

The general edits suggested above and the specific comments below constitute a major revision.

Abstract

The first sentence does not capture the main thrust of the research and what makes it new and exciting - the influence of hunting.

L 19: The data presented do not a causal relationship

Background

The authors touch on many different topics in the background – trophic cascades, non-linear dynamics, and human-predator conflicts. While these are all interesting considerations in the story, the section needs work on guiding the reader towards a central question. This section would benefit from pronounced focus on how adding human impacts on otters (and adding humans to ecosystem conceptions in general) can contribute towards our understanding. The connection made between urchins and sea otter-human conflicts is misleading as written (paragraph at L 42). Red urchins are not the only important recreational and commercial shellfish species who are otter prey. They are certainly not the most important per revenue or # of permits. A reader who doesn't know the system may think, based on the section written as-is, that the concerns over otters are all related to their consumption of urchins. Please clarify. Related to this – L58-60, I do not follow the logic to why this is the most important information need (compared to say the impact of otters on just shellfish). The research presented also includes kelp (an important foundation species, as mentioned) and hunting, so I suggest refocusing this paragraph. The authors mention the landscape of fear (and to the “risk landscape” in the Abstract), but this needs elaboration. While the data is not available to tease apart the direct vs indirect effects of hunting (numeric vs behavioral), the authors should do more to address the topic in both the background and discussion.

L 48-50: There are many ways that kelp benefit humans. Either add some more examples of ecosystem services or make the sentence more general.

L 53-54: "Limited" how? Spatially? Numerically? Please clarify.

L 56: Consider citing Carswell, L. P., S. G. Speckman, and V. A. Gill. 2015. Shellfish fishery conflicts and perceptions of sea otters in California and Alaska. Pages 333–368 in S. E. Larson, J. L. Bodkin, and G. R. VanBlaircom, editors. Sea otter conservation. Academic Press, Boston, Massachusetts, USA.

The last sentence of the section makes no mention of humans, and it should.

Methods

The methods appear sound and the authors do well pointing out the limitations of their dataset. If the approximate minimum distance of 1988 survey sites to Sitka is known, it should be included here. While the exact distributions of distances is unknown, any additional information on how 1988 survey sites were distributed with regards to distance from Sitka will be helpful for comparing to the later surveys.

Include if the boat was stationary/anchored during the otter surveys or if it was following a transect, grid, etc.

L 77: Citation needed for Torch Bay carrying capacity (as at line 93)

L 88-90: Is this inverse relationship from a citation or your own research?

Results

The results are thoroughly described, but some portions should be placed in the discussion. Some of the statements in this section are not solidly supported (see below).

"Non linear" (used in the discussion as well) doesn't seem like the right phrase to describe some the results presented on ecosystem state. A threshold is apparent, but insistence on "nonlinear" is misleading. Perhaps "chaotic" or "variable".

L 194-196: How did you calculate this number?

L 201: This should go in discussion, not results

Fig 2, panels for 2018 urchins and kelp - there are trends here, but low explanatory power. I see changes in variability over distance (which the authors rightly mention on L 251). L 318-319, 303-305 overstate the conclusions that can be drawn from this data.

Fig 3: Consider adding sea otter harvest numbers (L190-194) or density (Fig S2) to the Sitka side to really hammer the point home (the years won't match up, but just create a new x axis).

L 213/ Fig 4: R-square needed. As with figure 2, there seems to be low explanatory power

L224-227: This sentence is confusing, partially from the phrase "indication of an increase". I also do not see an "indication of an increase with distance" in Fig 2. Please re-write for clarity.

L237-238: Sentence should end at "small", The last phrase is for the discussion

L 242-243: This should go in discussion, not results

Fig S3: include Sitka on map

Discussion

The authors clearly describe the ecosystem changes over time and space as a result of sea otter recolonization and sea otter hunting. They highlight the importance of spatial scale when looking at ecosystem patterns, which is important for management considerations. While it is well-written overall, more could be done to highlight the unique aspects of this study (as mentioned before), to tell a cohesive narrative, and there are a few conclusions that need more support.

L 293-294: Please remove or provide a citation. The present research doesn't compare trophic cascades across systems.

L ~307-314: this section talks about how otters cannot coexist with productive shellfisheries, but the next paragraph is about how patchiness can allow for both to occur (and see Slade et al 201). Streamline between these two paragraphs. I re-emphasize that otter population thresholds and

per-capita effects on shellfish are beyond the scope of this study – these aspects should be removed from the discussion.

L 352-354: This thought seems incomplete.

Referee: 2

Comments to the Author(s).

General: I appreciate the efforts to revise the manuscript with respect to the sea star. However, although the "natural experiment" of a gradient in hunting pressure is a great addition to a well-established story, this aspect is as not well supported as others, which are more confirmatory of previous research, in spite of the enormous sampling efforts in the past and from another location. I feel that if it had been the focus, there would have been more effort here (e.g., sampling/data/local knowledge), and a more convincing case could have been made with less uncertainty with regards to nonlinearity and thresholds.

Specific comments

1. While I agree that there is overwhelming evidence for alternative community states (presented both here and in previous work) and that there must be some nonlinear relationships underlining it, I still do not see the data presented here being able to show a threshold value or suggest there are "threshold values in otter abundance" (lines 301 and 384). First, the otter abundance data are from one sampling (see additional comment and questions below) and were used to create "a spatial index" (line 104). These data are then shown as "probability of otter presence" in the figures, but how three successive counts of otter numbers are converted into probabilities is not given. Moreover, and more importantly, the link to the proposed threshold value of $K/2$ is not provided either (line 305). Perhaps I missed something here.
2. Otter sampling (lines 100-101): it is stated "We then repeated this sampling protocol two more times.", but it is not clear if this was done immediately (i.e., a single 5+5+5=15-minute sampling at each site) or on other days. If the latter, one would have more confidence in the abundance data. Indeed, more details are needed here (period – including dates – of sampling, how many days of sampling, time of day, weather, etc.) because if all the sampling was done over a short period, local conditions could influence the results.
3. Distance: I think the use of Euclidian distances is not ideal, and the shortest water route (passable by boats) is a more appropriate measure. Ideally, interviews with local hunters could have provided this kind of information, but even navigation charts would be better. This is especially true if the study focusses on the relationship between distance and other variables. The harbour from which hunter depart should be indicated on the map.
4. Statistics: again, I am no expert, but with the emphasis on nonlinearities, then some better assessment is need to compare them with simple linear models, perhaps segmented/threshold regression or some goodness-of-fit.
5. Torch Bay changes: self-thinning is invoked to explain declines in kelp density, but even without knowing the species well, I doubt that their longevity spans decades.
6. Nonlinearity: The statement "functional relationships among otters, urchins, and kelp were nonlinear, resulting in alternate community states" in the Abstract is overreaching -- see previous comments and above.

Author's Response to Decision Letter for (RSPB-2021-1697.R0)

See Appendix B.

RSPB-2021-1697.R1 (Revision)

Review form: Reviewer 1

Recommendation

Accept with minor revision (please list in comments)

Scientific importance: Is the manuscript an original and important contribution to its field?

Good

General interest: Is the paper of sufficient general interest?

Good

Quality of the paper: Is the overall quality of the paper suitable?

Good

Is the length of the paper justified?

Yes

Should the paper be seen by a specialist statistical reviewer?

No

Do you have any concerns about statistical analyses in this paper? If so, please specify them explicitly in your report.

No

It is a condition of publication that authors make their supporting data, code and materials available - either as supplementary material or hosted in an external repository. Please rate, if applicable, the supporting data on the following criteria.

Is it accessible?

Yes

Is it clear?

Yes

Is it adequate?

Yes

Do you have any ethical concerns with this paper?

No

Comments to the Author

The manuscript has been sufficiently improved. The authors have done considerable work to reframe the ms, focusing on the truly unique aspects of their study, placing it in context with recent studies and management concerns, and adequately addressing the methodological shortcomings.

A few minor comments:

L 22: region-wide (hyphen missing)

L 48: reductions in shellfish due to otters also occur in eelgrass habitats. Consider changing phrasing to "sea otter-driven trophic cascade".

L 54-66: It's not clear how this paragraph fits in with the study, as functional relationships in abundance were not addressed.

L 78: comma missing after "attenuation"

L 83: I suggest using either "Alaska Native" (as it is the first time this group is mentioned in the text and folks unfamiliar with AK may be unfamiliar with the official term) or "Indigenous"

Fig 2: specify that the non-linear relationships are square-root transformed

L328-330: I don't think this is impacting the outcome of the trophic cascade, as the cascade is still there but with an additional level.

Decision letter (RSPB-2021-1697.R1)

23-Nov-2021

Dear Dr Kroeker

I am pleased to inform you that your Review manuscript RSPB-2021-1697.R1 entitled "Southeast Alaskan kelp forests: inferences of process from large-scale patterns of variation in space and time" has been accepted for publication in Proceedings B.

The referee(s) do not recommend any further changes. Therefore, please proof-read your manuscript carefully and upload your final files for publication. Because the schedule for publication is very tight, it is a condition of publication that you submit the revised version of your manuscript within 7 days. If you do not think you will be able to meet this date please let me know immediately.

To upload your manuscript, log into <http://mc.manuscriptcentral.com/prsb> and enter your Author Centre, where you will find your manuscript title listed under "Manuscripts with Decisions." Under "Actions," click on "Create a Revision." Your manuscript number has been appended to denote a revision.

You will be unable to make your revisions on the originally submitted version of the manuscript. Instead, upload a new version through your Author Centre.

1) A text file of the manuscript (doc, txt, rtf or tex), including the references, tables (including captions) and figure captions. Please remove any tracked changes from the text before submission. PDF files are not an accepted format for the "Main Document".

2) A separate electronic file of each figure (tiff, EPS or print-quality PDF preferred). The format should be produced directly from original creation package, or original software format. Please note that PowerPoint files are not accepted.

3) Electronic supplementary material: this should be contained in a separate file from the main text and the file name should contain the author's name and journal name, e.g. `authorname_procb_ESM_figures.pdf`

All supplementary materials accompanying an accepted article will be treated as in their final form. They will be published alongside the paper on the journal website and posted on the online figshare repository. Files on figshare will be made available approximately one week before the

accompanying article so that the supplementary material can be attributed a unique DOI. Please see: <https://royalsociety.org/journals/authors/author-guidelines/>

4) Data-Sharing and data citation

It is a condition of publication that data supporting your paper are made available. Data should be made available either in the electronic supplementary material or through an appropriate repository. Details of how to access data should be included in your paper. Please see <https://royalsociety.org/journals/ethics-policies/data-sharing-mining/> for more details.

If you wish to submit your data to Dryad (<http://datadryad.org/>) and have not already done so you can submit your data via this link <http://datadryad.org/submit?journalID=RSPB&manu=RSPB-2021-1697.R1> which will take you to your unique entry in the Dryad repository.

Once again, thank you for submitting your manuscript to Proceedings B and I look forward to receiving your final version. If you have any questions at all, please do not hesitate to get in touch.

Sincerely,
Dr Daniel Costa
Editor, Proceedings B
<mailto:proceedingsb@royalsociety.org>

Reviewer(s)' Comments to Author:

Referee: 1

Comments to the Author(s)

The manuscript has been sufficiently improved. The authors have done considerable work to reframe the ms, focusing on the truly unique aspects of their study, placing it in context with recent studies and management concerns, and adequately addressing the methodological shortcomings.

A few minor comments:

L 22: region-wide (hyphen missing)

L 48: reductions in shellfish due to otters also occur in eelgrass habitats. Consider changing phrasing to "sea otter-driven trophic cascade".

L 54-66: It's not clear how this paragraph fits in with the study, as functional relationships in abundance were not addressed.

L 78: comma missing after "attenuation"

L 83: I suggest using either "Alaska Native" (as it is the first time this group is mentioned in the text and folks unfamiliar with AK may be unfamiliar with the official term) or "Indigenous"

Fig 2: specify that the non-linear relationships are square-root transformed

L328-330: I don't think this is impacting the outcome of the trophic cascade, as the cascade is still there but with an additional level.

Author's Response to Decision Letter for (RSPB-2021-1697.R1)

See Appendix C.

Decision letter (RSPB-2021-1697.R2)

13-Dec-2021

Dear Dr Kroeker

I am pleased to inform you that your manuscript entitled "Southeast Alaskan kelp forests: inferences of process from large-scale patterns of variation in space and time" has been accepted for publication in Proceedings B.

If you are likely to be away from e-mail contact please let us know. Due to rapid publication and an extremely tight schedule, if comments are not received, we may publish the paper as it stands. If you have any queries regarding the production of your final article or the publication date please contact procb_proofs@royalsociety.org

Data Accessibility section

Open Access

Paper charges

Sincerely,

Appendix A

Dear Editor,

Thank you for the opportunity to resubmit our manuscript to Proc B. We appreciated the feedback from the reviewers and Associate Editor, and we have revised our manuscript accordingly. In particular, we have refocused our framing on the effects of sea otter harvesting and spatial variability in sea otter density on kelp forest community structure, and limited our discussion of Pycnopodia and seastar wasting disease to a brief paragraph in the discussion. We address further changes in the detailed point-by-point response below.

*Sincerely,
Kristy Kroeker on behalf of all co-authors*

Associate Editor
Board Member: 1

Comments to Author:

We received two constructive reviews, both of which commend the study for the long-term scope of the data, particularly underwater data from cold marine habitat. I agree with those points, but both referees also highlighted that the scope and conclusions are a major over reach with regards to the role of sea stars in the system. The fact that there are no actual data or analysis of sea stars, that the seastar wasting disease event was synchronous across areas, and any spatial contrasts are limited between just two areas, puts substantial limitations on what can be inferred about sea stars. In particular there is no spatial 'control' or contrast area to compare the SSWD event relative to any other broad spatial scale change. Such is the limitation of analyzing data from a 'natural experiment', which is not a criticism of the fieldwork effort, but rather a reality that confounding variables cannot be accounted for via the temporal and spatial contrasts available from the data. I agree with the referees that these weaknesses make components of the paper about sea stars a major over reach of what can be supported by the data and analyses, and that at most, the sea star part of the story is a minor discussion point. With that in mind, the results remain interesting but represent a minor incremental and confirmatory contribution to an already well understood otter-urchin-kelp cascade. I'll also highlight the issue of confounding terminology of 'nonlinear' and tipping points that in this paper unfortunately perpetuates missuse and confusion. Yes it is true that some of the regressions are not straight lines, but that is not the same thing as nonlinear dynamics and the tipping points that can emerge due to bifurcations and unstable equilibria.

Thank you for these comments. We have removed the framing regarding seastars and limited this to a paragraph in the discussion. In addition, we have removed any mention of tipping points.

Reviewer(s)' Comments to Author:

Referee: 1

Comments to the Author(s)

This manuscript describes spatial and temporal patterns in kelp and urchin biomass in SE Alaska, and relate them to patterns in otter abundance. They show patterns consistent with the classic kelp-urchin-otter trophic cascade and suggest the ecosystem implications otter harvesting, even without extirpation. They found that in Sitka Sound (an area that became recolonized by an abundance of sea otters) patterns in otter presence over time, and over space in the most recent survey period, is positively related to kelp density and negatively related to urchin biomass. Additionally, they found that the algal community in Sitka Sound became more spatially heterogeneous over time. In Torch Bay, where otter abundance is low, they found high kelp biomass in all time periods, an increase in small urchins in the latest time period, and a changing algal community over time with no significant change in spatial community dissimilarity. The authors go on to explain these patterns via trophic cascade, but also make an attempt to relate them to the loss of Pycnopodia from Sea Star Wasting Disease.

Strong ecological baseline data such as this is valuable and rare. The role of trophic redundancy in shaping ecosystems is important from both a theory and management standpoint. However, without data on Pycnopodia over time, one cannot rule out alternate drivers, such as oceanographic changes from the Warm Blob. Per Konar et al 2019 and Harvell et al 2019, Pycno only became absent after SSWD in some areas, and could still persist as a significant part of the community. Reference to data on sea star abundance in the study area would be vital to bolster the related claims. The SSWD story is likely important here, but unfortunately lacks data. However, the authors present interesting data that suggests the ecosystem effects of otter harvest. In my opinion, the manuscript would be strengthened by a much stronger focus on that aspect of the story in the abstract, intro, and discussion. The resultant paper would be important from a management standpoint.

Thank you. We have taken your suggestion and refocused our framing on the harvesting and management story.

Additional comments:

Otter encounters increased with distance from Sitka, which the author's attribute to harvest patterns. Could this be related to other patterns (oceanographic, vessel traffic, etc)? Some background information of site fidelity in otters would also be helpful. Could the pattern of increasing otter encounter at more kelp-y locations be due to otter habitat preference?

The spatial patterns could ostensibly be related to vessel traffic as well, although we note that areas of high vessel traffic and human activity are used extensively by sea otters in California, with some of the highest sea otter densities (Elkhorn Slough and Monterey) corresponding to the areas of highest human activities. Moreover, elsewhere in southeast Alaska it has been found that areas of higher than average population

density and boat traffic are also preferred habitat for sea otters, based on a separate analysis currently in press in the journal "Movement Ecology" ("Diffusion modeling reveals effects of multiple release sites and human activity on a recolonizing apex predator", by Joseph Eisaguirre, Perry Williams, Xinyi Lu, Michelle Kissling, William Beatty, George Esslinger, Jamie Womble, Mevin Hooten, DOI: 10.21203/rs.3.rs-341528/v1). Thus, we believe that human activity by itself (excluding hunting) is unlikely to explain patterns of avoidance behavior.

The habitat preference hypothesis is harder to discount, in part because of the difficulty in establishing the temporal order of cause and effect. Specifically, we cannot definitively answer the question "is there less kelp now in the inner sound because sea otters moved away from this area (leading to increased urchin grazing, as we have documented here), OR did the kelp start to disappear first, causing sea otters to move away to areas with more reliable kelp beds?" In the absence of a more complete time series of information we cannot fully resolve this question, although we note that sea otters in many areas occur at high densities in habitats that are completely devoid of kelp. In fact, within SE Alaska the highest recorded densities of sea otters are found in Glacier Bay, in habitats with little or no kelp canopy cover.

We agree with the reviewer that more background information about sea otter movements, site fidelity and spatial ecology could be helpful for a reader in interpreting our results, and we have added in text to summarize published information on sea otter spatial ecology, with appropriate references.

Figs 2, 4 and associated results: what are the goodness of fit values for the regressions? These would help evaluate the effect of distance from town.

We have added r-squared values to all panels with a significant relationship between distance and the fitted y parameter.

Fig 3: an added panel of otter population over time would be useful here. Additionally, include indication where there is no data, so as to not confuse with the absence of one of the groups during the time period.

We have added a figure to the supplementary materials depicting trends in otter populations through time in both Torch Bay and Sitka Sound. In addition, we added NA for periods not sampled in Figure 3.

Comments to the Author(s)

General comments : This manuscript addresses interesting questions on the spatial and temporal dynamics of a keystone predator, building on a well-studied system, namely the sea otter-urchin-kelp trophic cascade in Alaska. The study spans an unusually long period of time (several decades), and although the spatial extent is less noteworthy (two areas), there is an interesting contrast between them and the sampling within each area is impressive considering the effort needed for underwater research, especially in

remote, cold waters.

Unfortunately, the main premise – an assessment of the relative importance of an alternative keystone predator, a sea star – is not supported by any data, and it is simply assumed that any impact of this mesopredator has been removed by the pandemic disease that has affected many sea star species in the northeast Pacific. While the use of such a natural experiment is not in itself a flaw, especially given the temporal scale of the study, there are limitations in the interpretation of the patterns observed. In particular, a comparison of just two areas is always a concern (i.e. no replication in the statistical sense) as they can differ in many ways. Likewise, temporal changes are difficult to interpret when they occur over the entire system of interest. This is not to say that I am not convinced that this study provides evidence for a minor role of the sea star, but I see this aspect more as a discussion point rather than as the main thrust of the manuscript. Also, given the inherent noise in natural systems, I am not certain how strong a sea-star signal would need to be detectable.

We agree and have limited our discussion of seastars as an observation to be considered, but it is no longer a primary thrust of the paper.

I am also not convinced that the relationships are sufficiently 'non-linear' to be described as a tipping point (see specific comments below).

We agree. We have removed mentions of tipping points.

Thus, overall, I found the work to add another chapter to a fascinating and well-developed story, but perhaps without remarkable novelty (I actually found the gradient of hunting pressure to be the more interesting aspect of the study). To attract a wider audience, I think the authors would need to balance the manuscript better, and if they do not have adequate data of their own, perhaps incorporate those from earlier studies to better justify the intriguing question of the relative importance of different predators.

Thank you. We have taken your suggestion and focused the manuscript on the gradient in hunting pressure and resulting patchiness in otter effects.

Specific comments:

Questions (line 35): references are needed to support this assertion, i.e., who has been asking these questions?

This statement is no longer applicable because the sentence has been removed.

Autotrophic kelp (line 48): all kelp are autotrophic, thus this adjective not needed.

Removed.

Sea star data (lines 77 & 78): this is all that is offered in terms of sea star population

changes. The references are either older (21) or about the disease (25) and the rest is unpublished or observational. Again, fine to support a discussion point, but not for “disentangling” (line 14) the relative effects of these two predators.

See comments about reframing and limiting our discussion of SSWD.

Sea otter densities: Similarly, the data for sea otter is limited – either all or none, an index taken at a single moment in time (Figure 2) or assumed to be constant (Torch Bay). Moreover, it is hard to compare the two areas – data are provided at a population level and not as true densities, e.g., otter/km² or km of shoreline (lines 172+).

We appreciate this point, and we have now provided the time series of sea otter densities in the Torch Bay subregion vs Sitka Sound subregion (density = the number of independent otters per km² of habitat, defined as the area of benthos between 0 and 40m depth) as a supplementary figure (Fig S2).

Hunter behavior (line 87): how is this known? Perhaps it is more of a reasonable assumption, but if that is the case, it should be stated as such.

We have added “ostensibly” to indicate this is an assumption.

Statistics: I am not an expert, but I think that considering Year to be a fixed factor (line 143) needs justification (usually a random factor). Moreover, treating location-year combinations as levels within a single factor (line 152) also seems unusual – typically a crossed design as least with traditional ANOVAs. This also needs support.

For the editor: We used year as a fixed effect rather than as a random effect as discussed by the reviewer because we were specifically interested in the potential pattern of change in the examined variables as a function of period. This allowed an examination of the potential effects of otters on the system (e.g., otters were not present in Sitka Sound in 1988, were at their peak in 2009 and were decreasing as a function of (probably) hunting by 2018).

With respect to the comment “Moreover, treating location-year combinations as levels within a single factor (line 152) also seems unusual” . This was done because the years sampled we not the same (1988, 2003, 2019 for Torch and 1988, 2009 and 2018 for Sitka Sound). Moreover we wanted to see how the communities differed over time at the two sites. Hence we used the 6 combinations as levels in a single factor.

Background: the information in lines 171-190 do not seem to be the results of this study and would be better presented in the Introduction or Discussion (or perhaps Methods).

This has been removed.

Distance (line 191): how was distance measured? Shortest water route, as the crow flies, or following the coast? Moreover, from Figure 1, there are clearly three close sites

(all approximately 2-3 km from Sitka as the crow flies) whereas the graphs in Figure 2 show only two close sites but both are nearly 4 km away.

Distance was modeled using simple Euclidean distances (as the crow flies). Because we were using distance as a metric for interactions with fishers and otters and because this is in an island dominated Sound it was unclear what other distance metric would be better, given that there are many ways to go from point A to B through a group of islands. With respect to the distances on the map vs the graph, the distances depicted on the graph is correct. We used the major harbors in Sitka as the start distance for measurements, so the point on the map indicates Sitka is south of the major harbors.

Urchin abundance (line 203): this trend seems “similar” only in being negative. The difference between the years is much more striking.

We have removed “similar”.

Sharp decline (line 206): I think that this is in the eye of beholder. There is certainly more scatter below (A) and above (B) the 0.5 value, but the variance is expected to increase as the mean does.

We have removed “sharp”.

Decreases (e.g., line 230): I find here and elsewhere the use of percentages and folds to describe decreases nonintuitive. Instead of saying there is a “1.5 fold decrease”, I suggest something along the lines of “decrease to 66% of previous levels”

We have changed these to % change throughout.

State space (lines 235+): I think that the idea of alternative stable states within this system is well established and the non-linearity here is simply driven by the variation at the endpoints.

We were unclear what the reviewer was asking us to do in this statement. Thus, we have edited this section to be more clear regarding our interpretation.

Variability in community structure (line 260): I am not certain what supports the statement that there the significant differences within locations – the size of the ellipses or their separation. Indeed, I find this figure to be of limited use, but that may be due being less familiar with multivariate analyses. Regardless, from this figure, it looks like Torch Bay has changed much more than Sitka Sound over time, which is not the message I have gotten from the other data and analyses. Perhaps this is being largely driven by changes in the species composition of the kelp assemblage, which is never discussed other than the mention on line 267.

We have added more background for interpreting the nMDS plot in this paragraph.

Duggin's observations (line 305): from what time period were these observations?

We have added the time period in the text. It was the late 70's and early 80's.

Combined effects/alternative explanations (lines 309-311): while I agree with this statement, I find it inconsistent with the claim that this study has "disentangled" the relative contributions of these three factors. Indeed, alternate explanations are numerous (lines 337-340 / lines 345-46), and again, the impact of sea stars is not resolved here as these alternatives can also explain the patterns observed. Indeed, the lack of kelp in Sitka Sound in 1988 is a strong suggestion that the sea star is not a keystone predator – the remaining observations in later years only reinforce this idea (i.e., the otter drives the show). The only evidence that I see otherwise is the 3x increase in abundance in Torch Bay from 2003 to 2018, but many other things could have changed over that 15-year period, and such changes are not uncommon just due to natural variation.

Thank you for your thoughts here. We have taken your suggestions and substantially changed the framing such that this comment is no longer applicable.

Coralline algae (Figure 6): these appear in this figure but are not mentioned elsewhere, in particular the methods – how were these quantified? If not density (difficult with encrusting organisms), then how can they be compared to the kelp? Likewise, *Codium* appears only in this figure.

Thanks for catching this. We have added a few sentences to the methods that outline our approach.

Minor comments

Biomass vs. biomass density: the justification for using estimates of biomass is fine, but the term "biomass density" is confusing and not consistently used throughout the manuscript (sometimes "biomass" sometime just "density" (e.g, line 201).

We have gone through and made sure that we use "biomass density" throughout.

Appendix B

Gorra et al. Response to Reviewers

Dear Dr. Costa,

Thank you for the opportunity to revise our manuscript and respond to the reviewers' comments. In addition to more minor changes that we highlight below, we have also made more substantial adjustments in our framing to better highlight the new insights that are gained from this study. In particular, we have focused on how including humans in the food web can provide insight into how sea otters, kelp forests, and shellfisheries can coexist, despite the well documented alternative stable states associated with sea otters in this region. We think our manuscript has greatly benefitted from the reviewer comments, and we look forward to your response.

Best,

Kristy Kroeker and colleagues

Referee: 1

Comments to the Author(s).

The manuscript deals with patterns of kelp, red urchin, and sea otters at two locations in Alaska over the past three decades – one where otters are hunted by humans, and one where they are not. The “classic” trophic cascade appears in the data, but with the added element of inferred hunting effects decreasing otter abundance closer to the town of Sitka. The research contributes to understanding the ecosystem dynamics associated with predator re-establishment and connections to human activities, which is an important management topic. The manuscript is well-written overall and most of the conclusions are well-supported, but it needs increased focus to form a cohesive narrative.

The impacts of sea otters on North Pacific kelp ecosystems is well-established. The authors should emphasize the new insight they bring to the table. The manuscript presents interesting data on the ecosystem effects of hunting sea otters - the main topic/question that the research addresses. However, that thread is lost at many points in the narrative. This topic is of recent interest – see the recent Slade et al. publication*. I think this citation emboldens the authors' claims and should be incorporated into the manuscript introduction and discussion.

*Slade, E., McKechnie, I. & Salomon, A.K. Archaeological and Contemporary Evidence Indicates Low Sea Otter Prevalence on the Pacific Northwest Coast During the Late Holocene. *Ecosystems* (2021). <https://doi.org/10.1007/s10021-021-00671-3>

Thank you for your thoughtful critique. We have taken your advice and better framed the contribution by focusing on how human harvesting can create small-scale spatial variability in kelp biomass within a region. We have added a paragraph to the introduction regarding what is and isn't known about human-otter interactions, and better threaded this theme throughout the introduction and discussion. Moreover, we have removed some of the framing around functional relationships and thresholds. The

Slade et al (2021) paper is now cited, among several others around this theme. Thank you so much for bringing it to our attention.

While the data provide increased understanding of the role of humans in shaping marine ecosystems, the implications of some of the results are overstated. There seems to be an underlying assertion that the manuscript deals with assessing otter population levels for management (lines 39-40, 305-314). This is a reach for the data presented and those sections should be re-written or removed.

Point taken. We have removed much of this discussion.

The general edits suggested above and the specific comments below constitute a major revision.

Abstract

The first sentence does not capture the main thrust of the research and what makes it new and exciting - the influence of hunting.

We have reframed the abstract, and the first two sentences now highlight the new insights gained by considering humans as part of the ecosystem.

L 19: The data presented do not a causal relationship

We have replaced “causal” with “is associated with.”

Background

The authors touch on many different topics in the background – trophic cascades, non-linear dynamics, and human-predator conflicts. While these are all interesting considerations in the story, the section needs work on guiding the reader towards a central question. This section would benefit from pronounced focus on how adding human impacts on otters (and adding humans to ecosystem conceptions in general) can contribute towards our understanding.

We have taken this to heart and emphasized this aspect of the work in the introduction.

The connection made between urchins and sea otter-human conflicts is misleading as written (paragraph at L 42). Red urchins are not the only important recreational and commercial shellfish species who are otter prey. They are certainly not the most important per revenue or # of permits. A reader who doesn't know the system may think, based on the section written as-is, that the concerns over otters are all related to their consumption of urchins. Please clarify.

We have clarified this throughout the introduction and discussion (Lines 71-72 “Sea otters are also voracious predators of other shellfish, including abalone, mussels, and clams (Singh et al. 2013, Lee et al. 2016, Hale et al. 2019)”).

Related to this – L58-60, I do not follow the logic to why this is the most important information need (compared to say the impact of otters on just shellfish). The research presented also includes kelp (an important foundation species, as mentioned) and hunting, so I suggest refocusing this paragraph.

We agree. Our research does not address the effects of otters on shellfisheries more broadly. Instead, we address whether all taxa within the trophic cascade (e.g., otters, urchins and kelps) can be maintained in a region. We have refined the language in this second paragraph to better highlight this (Lines 85-91).

The authors mention the landscape of fear (and to the “risk landscape” in the Abstract), but this needs elaboration. While the data is not available to tease apart the direct vs indirect effects of hunting (numeric vs behavioral), the authors should do more to address the topic in both the background and discussion.

We have added language to the abstract, introduction, and discussion, as well as a citation of landscapes of fear (Gaynor et al. 2019 [30]), to better describe what we mean by this phrase (Lines 24-27, 97-101, 358-361).

L 48-50: There are many ways that kelp benefit humans. Either add some more examples of ecosystem services or make the sentence more general.

We have changed this sentence to read, “In contrast, the positive indirect effects of sea otters on kelp commonly manifest as human benefits because kelp forests provide numerous ecosystem services, including habitat provisioning for other species, carbon sequestration, and wave attenuation among others.” (Lines 75-78)

L 53-54: “Limited” how? Spatially? Numerically? Please clarify.

We changed this wording to “...human harvest by coastal Native communities is allowed and has occurred in some areas (Lines 81-83).”

L 56: Consider citing Carswell, L. P., S. G. Speckman, and V. A. Gill. 2015. Shellfish fishery conflicts and perceptions of sea otters in California and Alaska. Pages 333–368 in S. E. Larson, J. L. Bodkin, and G. R. VanBlaircom, editors. Sea otter conservation. Academic Press, Boston, Massachusetts, USA.

Cited.

The last sentence of the section makes no mention of humans, and it should.

We have reworked this last paragraph, but the last sentence now reads, “Here, we use time series in two regions of SE Alaska spanning three decades to highlight the functional relationships between humans, sea otters, urchins and kelp created by within-region spatial variability in otter populations (Lines 101-103).”

Methods

The methods appear sound and the authors do well pointing out the limitations of their dataset.

Thank you.

If the approximate minimum distance of 1988 survey sites to Sitka is known, it should be included here. While the exact distributions of distances is unknown, any additional information on how 1988 survey sites were distributed with regards to distance from Sitka will be helpful for comparing to the later surveys.

The locations for the 2018 and 2019 sampling efforts were based on digital files with site numbers from 1988 and 2003/2009, as well as GPS locations for sites in 2003/2009. We erroneously thought the same numbers assigned to the sites in the first two sampling periods indicated they were the same site. We found out after sampling in 2018/2109 that the same numbers did not mean they were the same site. Prior to this revision, we had been unable to locate any maps or coordinates for the 1988 sampling, despite lots of effort! Fortuitously, we were able to locate the data in some very old files in the past couple of weeks. We have updated the text and figures accordingly to include the distance analyses in 1988 based on rough locations extracted from a hand-drawn map using Google Earth. It should be noted that the original sampling locations from 1988 are slightly farther from the town of Sitka than in subsequent sampling periods, although there is some overlap. The new data do not change the interpretation or story – as there is no relationship between urchin or kelp biomass density and distance from the town of Sitka. Urchin biomass is high and kelp density is low across all of the sites in 1988. We have added text to the methods (Lines 153-161) and new panels in Figure 2 and Figure S1.

Include if the boat was stationary/anchored during the otter surveys or if it was following a transect, grid, etc.

The boat was anchored. We have added this detail to the text (Line 136).

L 77: Citation needed for Torch Bay carrying capacity (as at line 93)

Added.

L 88-90: Is this inverse relationship from a citation or your own research?

This was from Raymond et al. (2019). We have clarified this in the text (Lines 123-125: Although exact harvest locations were not reported, the hunters did report that they endeavored to minimize their travel distances, resulting in an inverse relationship between harvest intensity and distance from population centers of hunters (Raymond et al. 2019)).

Results

The results are thoroughly described, but some portions should be placed in the discussion. Some of the statements in this section are not solidly supported (see below).

See responses to particular points below.

“Non linear” (used in the discussion as well) doesn’t seem like the right phrase to describe some the results presented on ecosystem state. A threshold is apparent, but insistence on “nonlinear” is misleading. Perhaps “chaotic” or “variable”.

We have removed discussions of non-linearities and now focus on the functional relationships and variability/patchiness within the region.

L 194-196: How did you calculate this number?

This was taken inferred from the analyses of Raymond et al. (2019). We have refined the text and added a citation (Lines 241-242).

L 201: This should go in discussion, not results

Removed.

Fig 2, panels for 2018 urchins and kelp - there are trends here, but low explanatory power. I see changes in variability over distance (which the authors rightly mention on L 251). L 318-319, 303-305 overstate the conclusions that can be drawn from this data.

We have removed the discussion of the transition point and refined the interpretation previously in Lines 318-319 (now Lines 346-348 “We found the sites with the fewest urchins were farther from the town of Sitka, whereas the sites with the least kelp were closest to the town – although there was some important variability in this relationship (Fig.S3). In particular, we found some intermediate states (with some urchins and low kelp density) scattered throughout the region. While it is unclear whether the intermediate states are at equilibria (or in the process of changing states), the overall patchiness in the density of urchins and kelp in 2018 indicates that both ecosystem states can co-occur when the presence or relative density of otters is patchy”).

Fig 3: Consider adding sea otter harvest numbers (L190-194) or density (Fig S2) to the Sitka side to really hammer the point home (the years won’t match up, but just create a new x axis).

We have added this to Fig 3.

L 213/Fig 4: R-square needed. As with figure 2, there seems to be low explanatory power

We have added R-square to Fig 4.

L224-227: This sentence is confusing, partially from the phrase “indication of an increase”. I also do not see an “indication of an increase with distance” in Fig 2. Please re-write for clarity.

We have removed this clause.

L237-238: Sentence should end at “small”, The last phrase is for the discussion

We have removed this clause as well.

L 242-243: This should go in discussion, not results

Removed.

Fig S3: include Sitka on map

Done.

Discussion

The authors clearly describe the ecosystem changes over time and space as a result of sea otter recolonization and sea otter hunting. They highlight the importance of spatial scale when looking at ecosystem patterns, which is important for management considerations. While it is well-written overall, more could be done to highlight the unique aspects of this study (as mentioned before), to tell a cohesive narrative, and there are a few conclusions that need more support.

We have substantially changed the framing and discussion per your recommendations.

L 293-294: Please remove or provide a citation. The present research doesn't compare trophic cascades across systems.

Removed.

L ~307-314: this section talks about how otters cannot coexist with productive shellfisheries, but the next paragraph is about how patchiness can allow for both to occur (and see Slade et al 201). Streamline between these two paragraphs. I re-emphasize that otter population thresholds and per-capita effects on shellfish are beyond the scope of this study – these aspects should be removed from the discussion.

We have removed the discussion of per capita effects on shellfish, and refocused these paragraphs substantially.

L 352-354: This thought seems incomplete.

We have rewritten this sentence for clarity (Lines 384-387 “Recent studies from other localities [28, 30] suggest sunflower stars can affect the distribution and abundance of urchins and kelp, and it is possible that SSWD contributed to shifts in community structure seen across both Torch Bay and Sitka Sound”).

Referee: 2

Comments to the Author(s).

General: I appreciate the efforts to revise the manuscript with respect to the sea star. However, although the "natural experiment" of a gradient in hunting pressure is a great addition to a well-established story, this aspect is as not well supported as others, which are more confirmatory of previous research, in spite of the enormous sampling efforts in the past and from another location. I feel that if it had been the focus, there would have been more effort here (e.g., sampling/data/local knowledge), and a more convincing case could have been made with less uncertainty with regards to nonlinearity and thresholds.

We have removed the emphasis and discussion regarding non-linearities and thresholds and instead have focused on the patchiness within a region associated with otter occupancy and purported human harvest. Unfortunately, it is very difficult to get information on human harvesting patterns for a variety of reasons, but the spatial patterns in otter occupancy that we report here are in agreement with human harvest of sea otters modeled based on geo-located data reported by hunters in Raymond et al. (2019). We have added additional text in reference to these previous findings (Lines 123-125 “Although exact harvest locations were not reported, the hunters did report that they endeavored to minimize their travel distances, resulting in an inverse relationship between harvest intensity and distance from population centers of hunters (Raymond et al. 2019).”).

Specific comments

1. While I agree that there is overwhelming evidence for alternative community states (presented both here and in previous work) and that there must be some nonlinear relationships underlining it, I still do not see the data presented here being able to show a threshold value or suggest there are “threshold values in otter abundance” (lines 301 and 384). First, the otter abundance data are from one sampling (see additional comment and questions below) and were used to create “a spatial index” (line 104). These data are then shown as “probability of otter presence” in the figures, but how three successive counts of otter numbers are converted into probabilities is not given.

We have added this text to clarify the methods used, “The spatial index was developed from a logistic regression using latitude and longitude as predictor variables and otter presence (1) or absence (0) as the response. Hence the fitted surface represented the probability of the presence of at least one otter as a function of location of geographic location (Lines 143-146).”

Moreover, and more importantly, the link to the proposed threshold value of $K/2$ is not provided either (line 305). Perhaps I missed something here.

We have removed the discussion of thresholds.

2. Otter sampling (lines 100-101): it is stated “We then repeated this sampling protocol two more times.”, but it is not clear if this was done immediately (i.e., a single 5+5+5=15-minute sampling at each site) or on other days. If the latter, one would have more confidence in the abundance data. Indeed, more details are needed here (period – including dates – of sampling, how many days of sampling, time of day, weather, etc.) because if all the sampling was done over a short period, local conditions could influence the results.

All three surveys at a site were done on one day, with the site surveys spread over a 10-day window. We have added this detail and the others requested in the methods (Lines 135-141). We recognize that it would be valuable to have more surveys for otters, but want to emphasize that the results from our surveys align with the analysis performed by Raymond et al. (2019) based on reported otter harvests in Sitka Sound. Using geo-located hunting data that was collected by USFWS taggers, who record information provided by hunters, Raymond et al. (2019) found a significant, linear effect of distance from the common hunter population centers in Sitka on otter harvest rates.

3. Distance: I think the use of Euclidian distances is not ideal, and the shortest water route (passable by boats) is a more appropriate measure. Ideally, interviews with local hunters could have provided this kind of information, but even navigation charts would be better. This is especially true if the study focusses on the relationship between distance and other variables. The harbour from which hunter depart should be indicated on the map.

This seems at first glance as a reasonable thing to do – but we thought about this and there are three harbors and depending on conditions (wind, swell, tide) routes differ between harbors and destinations, even in the same destination – so we used the midpoint of the harbors as the starting location and then calculated Euclidean distance to the destination as an unbiased estimate. We have clarified this in the text (Lines 208-211) and emphasize again that a similar analysis undertaken by Raymond et al. (2019) provides support for this approach. Moreover, because Sitka Sound is a fairly open and relatively protected embayment, without major navigational barriers, Euclidean distance at this scale provides a fairly good approximation to navigational distances for hunters and is simpler than a more elaborate least-cost-path analysis.

4. Statistics: again, I am no expert, but with the emphasis on nonlinearities, then some better assessment is need to compare them with simple linear models, perhaps segmented/threshold regression or some goodness-of-fit.

We have removed the focus on non-linearities based on feedback from this reviewer and others. However, we now provide a table in the supplementary materials that does compare the different model fits (Table S1).

5. Torch Bay changes: self-thinning is invoked to explain declines in kelp density, but even without knowing the species well, I doubt that their longevity spans decades.

We realize the use of “self-thinning” was confusing, and have replaced this with wording with those of “succession” (Lines 397-403). Macroalgae can recruit in very high numbers when space opens up and grazing pressure is lifted, and the counts used for density include both recruits and adult thalli. The reduction in thalli counts was most likely associated with longer-term succession processes whereby a kelp forest with mature kelp plants does not sustain the same number of recruits/individuals.

6. Nonlinearity: The statement "functional relationships among otters, urchins, and kelp were nonlinear, resulting in alternate community states" in the Abstract is overreaching -- see previous comments and above.

We have removed the mention of non-linearity in the abstract and elsewhere.

Appendix C

Dear Editor,

Please find the minor changes we have made in response to the comments from the reviewer's on the last round.

Kristy

--

L 22: region-wide (hyphen missing)

Changed.

L 48: reductions in shellfish due to otters also occur in eelgrass habitats. Consider changing phrasing to "sea otter-driven trophic cascade".

Changed.

L 54-66: It's not clear how this paragraph fits in with the study, as functional relationships in abundance were not addressed.

We have removed this paragraph.

L 78: comma missing after "attenuation"

Comma added.

L 83: I suggest using either "Alaska Native" (as it is the first time this group is mentioned in the text and folks unfamiliar with AK may be unfamiliar with the official term) or "Indigenous"

Changed to "indigenous".

Fig 2: specify that the non-linear relationships are square-root transformed

We have specified this in the legend for Figure 2.

L328-330: I don't think this is impacting the outcome of the trophic cascade, as the cascade is still there but with an additional level.

We have changed "impacted" to "mediated".